**communications** engineering

# Inverse-designed large field-of-view polychromatic metalens for tri-color scanning fiber endoscopy

Ningzhi Xie [1], Zhihao Zhou[1], Johannes E. Fröch [1,2], Matthew D. Carson[3], Arka Majumdar [1,2] ✉, Eric J. Seibel[3] & Karl F. Böhringer[1,4,5]

Metalenses, with their ultrathin thicknesses and their ease for achieving ultra small diameters, offer a promising alternative to refractive lenses in miniaturized imaging systems, such as endoscopes, potentially enabling applications in tightly confined spaces. However, traditional metalenses suffer from strong chromatic aberrations, limiting their utility in multi-color imaging. To address this limitation, here we present an inverse-designed polychromatic metalens with a diameter of 680 µm, focal length of 400 µm, and low dispersion across 3 distinct wavelengths at 643 nm, 532 nm, and 444 nm. The metalens collimates and steers light emitted from a scanning fiber tip, generating scanning beams across a 70° field-of-view to provide illumination for a scan-based imaging. The metalens provides a close-to-diffraction-limited 0.5° angular resolution, only restricted by the effective aperture of the system. The average relative efficiency among three design wavelengths is around 32% for on-axis angle and 13% averaged across the entire field-of-view. This work holds promise for the application of metalenses in endoscopes and other miniaturized imaging systems.

The endoscope, a medical imaging device to visually examine the internal organs of the human body, is widely utilized in disease diagnosis and visually guided surgery[1]. Recent advancements have extended endoscopic applications towards more confined spaces such as blood vessels[2–4], the brain, and the spinal cord[5,6]. These advancements necessitate further size reduction towards extreme dimensions as small as a few hundred microns in diameter and rigid tip length. A stringent limitation in current state-of-the-art endoscopes is imposed by their aperture diameter, thickness, and focal distance of the optical element, which are required to focus light either onto a sensor, into a fiber, or towards the target. In pursuit of their miniaturization, metalenses have emerged as a promising solution. These lenses consist of quasi-periodic arrays of sub-wavelength scatterers known as meta-atoms. Meta-atoms induce spatially varying phase modulations on incoming light, effectively manipulating the optical wavefront[7]. The thickness of a metalens, defined solely by the height of its constituent meta-atoms, is on the order of the wavelength. The diameter of a metalens can be easily scaled down sub-milimeter or even tens of micrometers, due to its fabrication through two-dimensional lithography. These distinctive attributes position the metalens as a promising candidate for supplanting traditional refractive lenses in optical imaging systems, presenting a compelling avenue for system miniaturization[8]. Leveraging their ultra-thin thickness and the

ease of aperture reduction, metalenses have been integrated into various kinds of endoscopic systems, including coherent fiber bundle endoscopes[9,10], fluorescence confocal endoscopes[11–14], optical coherence tomographic endoscopes[15], and scanning fiber endoscopes[16].

However, a metalens exhibits a pronounced dispersion in wavelength, akin to other diffractive lenses, where the focal length is inversely proportional to the wavelength[17]. This substantial dispersion introduces severe chromatic aberrations, impeding the adoption of metalenses in multi-color imaging applications. To eliminate chromatic aberrations, the meta-atoms are often strategically engineered, ensuring that the phase imparted by these elements scales linearly with the wavelength[18–22]. With these meta-atoms, it is possible to realize the phase profiles that can simultaneously focus the light at all wavelengths within the working wavelength range onto the focal plane. However, this approach is limited to metalenses with an unwrapped phase profile (having only one Fresnel zone), where the product of the numerical aperture (NA) and the lens diameter falls within the same order of magnitude as the wavelength. Lenses with such minute apertures could find some applications when directly integrated with single mode fibers[23]. However, for most imaging endoscopes, a large-aperture metalens featuring a wrapped phase profile is necessary. As the phase exhibits $2\pi$ jumps at specific wrapping points, achieving dispersionless behavior becomes

[1]Department of Electrical and Computer Engineering, University of Washington, Seattle, WA, USA. [2]Department of Physics, Seattle, Washington, USA. [3]Human Photonics Lab, Department of Mechanical Engineering, University of Washington, Seattle, WA, USA. [4]Department of Bioengineering, University of Washington, Seattle, WA, USA. [5]Institute for Nano-Engineered Systems, University of Washington, Seattle, WA, USA. ✉e-mail: arka@uw.edu

challenging. To achieve dispersionless behavior, the phase wrapping points need to be wavelength-dependent, preventing simultaneous satisfaction of phase profiles across a broad wavelength range through meta-atom dispersion engineering. While an alternative method for mitigating the effect of chromatic aberrations of metalenses using extended depth-of-focus has been demonstrated[24], this approach comes at the expense of compromising lens resolution and need for computational reconstruction. The quest for a diffraction-limited, large-aperture, achromatic metalens capable of operating across a wide wavelength spectrum remains an ongoing challenge.

For a wide range of applications, light at multiple distinct wavelengths (e.g., from a laser) is used for imaging instead of broadband spectral light from the environment. In this case the imperative for a dispersionless metalens spanning a continuous wavelength band can be relaxed. A metalens that maintains identical optical functionality for only multiple distinct wavelengths, known as the polychromatic metalens, can satisfy the requirement of these applications. Scanning fiber endoscopes (SFE) present such an application: a lens collimates and steers laser light from a scanning fiber tip to illuminate a scene. The scattered light is captured by a returning fiber to form an image[25]. By using monochromatic light at red, green, and blue (RGB) wavelengths, color information of the scene can be captured. Thus, by ensuring that the metalens operates only at those three wavelengths, we can create a color RGB-SFE. As a polychromatic metalens only needs to satisfy the phase profile for a select set of distinct wavelengths, the design challenges are more manageable than those of a broadband metalens. Polychromatic metalenses with large diameters have already been demonstrated by many groups via selecting meta-atoms from a large library of meta-atoms with diverse phase distributions[26–29]. An inverse design framework has been employed to design polychromatic metalenses for thermal imaging[30], coherent fiber bundle endoscopy[9], and augment reality display[31]. However, these polychromatic metalenses only work for near-axis imaging. Large field of view (FOV) metalens can be realized by using an aperture stop to spatially separate the incident light from different angles[32–34].

In this paper, via adopting the inverse design framework and the design principle of spatially separating light from different angles, we report a large FOV polychromatic metalens as a beam steering lens that can support ~ 70° FOV imaging in a RGB-SFE system, achieving a closed-to-diffraction-limits angular resolution of 0. 5°, and an average efficiency of 13% across the entire field-of-view.

## Results and discussion
### Inverse design of the RGB polychromatic metalens for SFE
In an SFE, the illuminating beam is achieved by resonantly scanning a single mode fiber in the focal plane of a projecting metalens, as illustrated in Fig. 1a. Specifically, the fiber tip position $(x, y)$ follows a spiral trajectory, whereas the light emitted from the fiber tip at different positions is collimated by different sections of the metalens and steered to distinct angles. Collimation and steering of the beam can be equivalently viewed as the metalens forming a projection of the scanning fiber tip to the infinitely distant far field. As the scanning beam sequentially illuminates different parts of a target object, the backscattered light is collected with a returning fiber and the signal is subsequently utilized to reconstruct the image of the target scene.

To approach this design challenge, which minimizes the divergence of the steered beams for three wavelengths, we first optimize the optical system in ray-tracing software. In simulation, we obtain the position and area of the incident beam on the metalens and the corresponding steering angle of the beam (relative to the optical axis), $\theta$, as a function of the fiber tip's lateral displacement. This defines the optical functionality of the ideal optics and sets the goal for the subsequent inverse design.

Figure 1b shows the ray-tracing simulation of the beam steering system utilizing a metalens. Light emitted from the fiber tip at different locations is collimated by different sections of the metalens and steered towards distinct directions in the far field. As the optical system is rotationally symmetric around the optical axis (z-axis), we only consider the fiber tip moving along the y-axis (the tangential axis) for optimization. The x-axis thus becomes the sagittal axis. We denote $y$ as the lateral displacement of the fiber tip, where

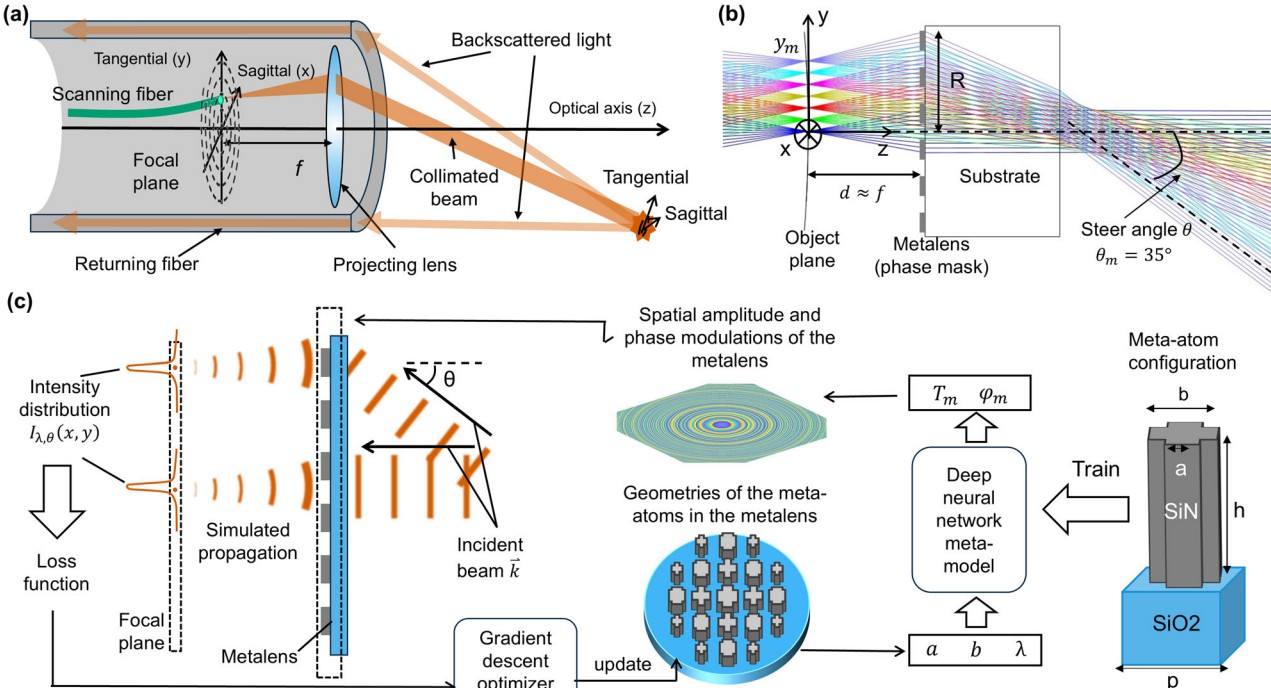

**Fig. 1 | The design of a polychromatic metalens for a scanning fiber endoscope (SFE) system. a** Sketch of the SFE system. **b** Cross-section (yz plane) of a ray-tracing simulation for the SFE beam steering system. $d \approx f = 0.4$ mm, $y_m = 0.4$ mm, $R = 0.34$ mm, $\theta_{max} = 35°$. **c** Inverse design approach of the RGB polychromatic metalenses as the projecting lens in the SFE. The positions and incident angles $\theta$ of the

incident beams are obtained by reversing the propagation direction of the steered beams in (b), the meta-atoms are $Si_3N_4$ on $SiO_2$ cross-shape structures with two tunable parameters $a$ and $b$, pitch $p = 300$ nm, height $h = 750$ nm. The working wavelengths $\lambda$ are 444 nm, 532 nm, and 643 nm. The output of the meta-model $T_m$ and $\varphi_m$ is the transmission and phase response of the meta-atoms.

the maximal displacement is $y_{max} = 0.24$ mm. In addition, the axial distance from the metalens to the fiber tip is $d = 0.40$ mm. The phase profile of the metalens is optimized as an even polynomial:

$$\psi(\rho) = a_1 \left(\frac{\rho}{R}\right)^2 + a_2 \left(\frac{\rho}{R}\right)^4 + a_3 \left(\frac{\rho}{R}\right)^6 \qquad (1)$$

Here, $\lambda$ is the wavelength of the light, $\rho$ is the distance from the lens center to a field point of the aperture, $R$ is the semi-diameter of the lens, which is set as $R = 0.34$ mm, and $a_1$, $a_2$, $a_3$ are optimization parameters. The optimized phase profiles satisfy $a_1 \gg a_2, a_3$, which indicates a parabolic type lens, with $a_1 = -\frac{\pi R^2}{f\lambda}$, and the focal length $f$ of the metalens approximately satisfying $f = d$. With such a metalens, the beam steering angle $\theta$ closely follows $\theta = \arcsin(y/d)$, with $\theta_{max} = \arcsin(y_{max}/d) = 35°$ defining a $2\theta_{max} = 70°$ full angular FOV of the SFE system. Details of the ray-tracing optimization can be found in the method section, and further aberration analysis can be found in the Supplementary Note 1.

With the phase profile obtained from the ray-tracing simulation, a monochromatic metalens can then be designed by mapping the phase response of meta-atoms to the phase profile wrapped by $2\pi$. However, as we require polychromatic functionality, the device must closely satisfy three different wrapped phase profiles for red (643 nm), green (532 nm), and blue (444 nm) wavelengths. These phase profiles have distinct $2\pi$ discontinuity points, precluding straightforward meta-atom to phase mapping. To achieve simultaneous perfect mapping for three different wrapped phase profiles, the possible phase responses of the meta-atoms, noted as $(\psi_R, \psi_G, \psi_B)$, would have to fully cover the entire $(0, 2\pi) \times (0, 2\pi) \times (0, 2\pi)$ range[17,26]. This requirement is impossible to achieve with simple meta-atom geometry, because the phase response at different wavelengths is then proportional to the meta-atom footprint. To alleviate this constraint, we use a cross-shaped meta-atom library. Compared to simple square or cylindrical shapes, this cross shape meta-atom diversifies the attainable phase range for the three wavelengths as indicated by previous works[30]. Compared to other more complicated shapes, the cross shape is still simple enough to be reliably fabricated. Although this meta-atom type has a limited spread of phase responses, we achieve sufficient coverage in the three-dimensional phase value range to closely implement the desired functionality of a RGB polychromatic metalens. Using these meta-atoms, we first choose a meta-atom at every local site of the metalens by minimizing the phase error between the desired and the achievable phase distribution for all three wavelengths. However, this local forward optimization does not necessary lead to the overall good performance of the metalens, as the light intensity distribution on the focal plane is cohesively affected by the phase and amplitude of all the meta-atoms. Therefore, we implement an inverse design framework, as shown in Fig. 1c, to concurrently optimize the geometric parameters of all meta-atoms at the desired wavelengths. The forward optimization design is used as the initial condition in the inverse design. The inverse design objective is to maximize the collimated output for the desired angular range. This optimization is performed using a gradient decent method to minimize a loss function that evaluates the performance of the metalens. Further details on the inverse design method are described in the method section.

## Simulated properties of the designed metalens

We utilize the angular spectrum method based on scalar diffraction theory to simulate the performance of the designed polychromatic metalens. First we analyze the beam intensity distribution in the optical (yz) plane, defined by the optical axis and the tangential axis. Light emitted from the fiber tip has a Gaussian distribution with a divergence angle defined by the numerical aperture (NA) of the fiber, which is set to 0.1 to match the NA of the single-mode fiber we use in later experiments:

$$\tan \alpha = \frac{\lambda}{\pi w_0} = NA = 0.1 \qquad (2)$$

where $\alpha$ is the divergence angle of the Gaussian beam and $w_0$ is the waist width.

The intensity distributions for different wavelengths and steering angles in the optical plane are summarized in Fig. 2a. Notably, the beams after transmitting the metalens are collimated, propagating along the optical (z) axis for the fiber tip at the center position, or being steered to the same angle $\theta = \arcsin y/f = 17.5°$ when the fiber tip is placed at $y = 120$ μm. This shows negligible dispersion in the system.

We further analyzed the angular intensity distribution in the transverse direction (i.e., perpendicular to the optical axis), which essentially determines the imaging quality of the SFE, as plotted in Fig. 2b, c. These angular distributions are calculated using the intensity distributions in k-space, which is obtained via performing a two-dimensional Fourier transform of the light field after transmission through the metalens. As can be seen, the steered beams have a slightly diverging angle due to their finite spatial extent, resulting in angular spreads of energy at the far field (the Fresnel factor $F = r_b^2/\lambda L \ll 1$, where $r_b = f \cdot NA$ is the radius of the incident beam on the metalens, $\lambda$ is the wavelength, and $L$ is the propagation distance of the steered beam). However, the beam is mostly collimated, evident by the high intensity in a confined angular section (normalized intensity > 0.1).

To better characterize the system, we define two quantitative figures-of-merit: the angular radius of the steered beams and the relative beam collimating efficiency. The beam angular tangential (sagittal) radius is defined as the full-width-at-half-maximum (FWHM) of the one-dimensional cross-section of the angular intensity distribution along the tangential (sagittal) axis. Beyond this radius the intensity drops to less than 10% of the maximum, effectively gauging the angular resolution of the SFE image. We calculated the steered beams' angular radii from the simulated angular intensity distribution, and compared them with that from a diffraction limited lens, as shown in Fig. 2d and e. The calculation of the diffraction limited steered beams' angular radii can be found in Supplementary Note 5.

As can be seen in Fig. 2d, e, the calculated angular radius as a function of steering angle matches closely with the diffraction limit. These beam angular radii are proportional to the wavelengths, and the tangential radii are inversely proportional to $\cos\theta$. These results are as expected from the angular resolutions of a typical optical imaging system, which are $\sim\lambda/D$. Here $D$ is the effective aperture with its value along the tangential axis being proportional to the cosine of the off-axis angle. With an incident beam diameter (effective aperture for the on-axis beam) of 80 μm on the metalens and across the designed 35° steering angle range, the steered beam has a diffraction-limited angular radius of under 0.45°, 0.38°, 0.32° for red, green, and blue wavelengths. This could be improved by increasing the NA of the fiber and thus the effective aperture.

We also note that in the simulated angular intensity distributions of the steered beams in Fig. 2b, there exist sidelobe spots with lower intensity but larger sizes compared to the collimated beam spot. Only a certain part of the total energy is concentrated within the collimated beam spot. Thus we define the relative collimating efficiency of our RGB metalens, which is the energy concentrated within the diffraction-limited angular radius of the steered beam divided by the total energy transmitting through the metalens. This efficiency will affect the imaging contrast in SFE. Figure 2f plots the simulated relative collimating efficiency of the designed metalens for 3 design wavelengths at different beam steering angles. In the simulation, 3 design wavelengths show similar efficiencies of around 50% at 0° steering angles, and the efficiencies decrease with increasing steering angles, down to around 25% at the maximal design steering angle of 35°. These efficiencies could be further improved to ~ 60% at 0° steering angles and ~ 40% at 35° by using meta-atoms with larger height or higher material index, which provides a better phase diversity, as can be seen in the simulation results of metalenses using 1500 nm tall $Si_3N_4$ or 750 nm tall $TiO_2$ in Fig. S7 in the Supplementary Note 6.

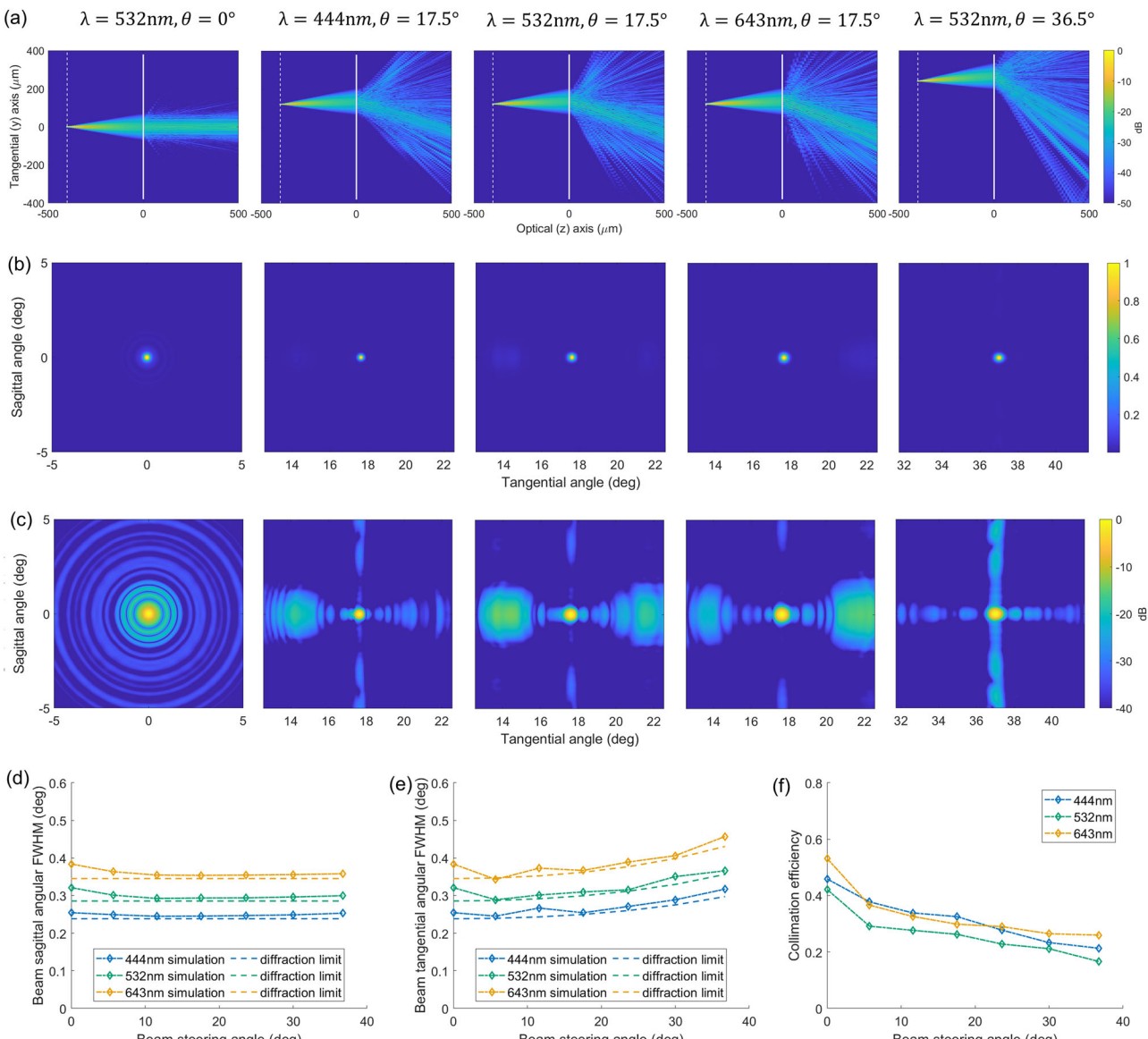

**Fig. 2 | Simulation performance of the metalens. a** Simulated beam intensity distributions in the yz plane for different wavelengths $\lambda$ and beam steer angles $\theta$. The emitted beams are Gaussian beams with a divergence angle defined by the NA = 0.1 of the fiber. Simulated angular intensity distribution of the steered beams for different $\lambda$ and $\theta$ at the far field. These distributions are normalized by the maximum intensity, plotted on a linear scale in (**b**) and on log scale in (**c**). The simulated and diffraction limited angular radius of the steered beams in sagittal (**d**) and tangential (**e**) direction at the far field as a function of the beam steer angle. For calculating the diffraction limits, the steered beam is assumed to be a Gaussian beam whose waist position is at the metalens and the amplitude profile at the waist equals the incident beam on the metalens. **f** The relative collimating efficiency as a function of the beam steer angle. The data is calculated from the numerical simulation.

## Characterization of the metalens

The metalens is fabricated using electron beam lithography followed by lift-off process and reactive ion etching (details are in the method section). Figure 3a, b shows scanning electron microscope images of the fabricated metalens.

We experimentally measured the angular intensity distribution of the steered beam at various steering angles using the setup shown in Fig. 3c. A single-mode fiber with an NA = 0.1 was placed with its tip fixed on a mechanical stage and adjustable within the focal plane of the metalens. The fiber is connected to laser sources (see the method section for details). A sensor was mounted on a rotational arm, whose rotation axis is co-axially aligned with the meta-optic center. Thus, as the tip of the single-mode fiber is parallelly translated along the tangential axis, the light coupled from the fiber tip is steered by the meta-optics, essentially emulating the scanning mechanism along the tangential axis. While for the scanning fiber tip, the tip positions have offset from the $z = 0$ plane,

and the emitted beams form angles relative to the optical axis instead of simply shifting parallelly along the tangential axis, the position offsets and the beam emitting angles are relatively small compared to the focal lengths of the metalens and the beam steering angle after transmitting through the metalens, as can be seen in the Table S1 in the Supplementary Note 1. We found in the simulation that these small differences between the parallelly shifted fiber tip and the scanning fiber tip do not lead to substantial differences in the intensity distributions of the steered beams. The rotation arm with sensor is then adjusted with respect to the sagittal axis to capture the steered beams at different angles. The distance from the light sensor to the metalens was fixed at $L = 80$ mm to satisfy the far field requirement (Fresnel factor $F = a^2/(\lambda L) = 0.18$, where $a = 2f \times$ NA = 80 μm is the diameter of the incident beam on the metalens, and $\lambda = 440$ nm is set as the shortest wavelength).

Figure 3d shows the angular intensity distribution for beams steered to 0° and 15° for the three different wavelengths on a linear scale. Importantly,

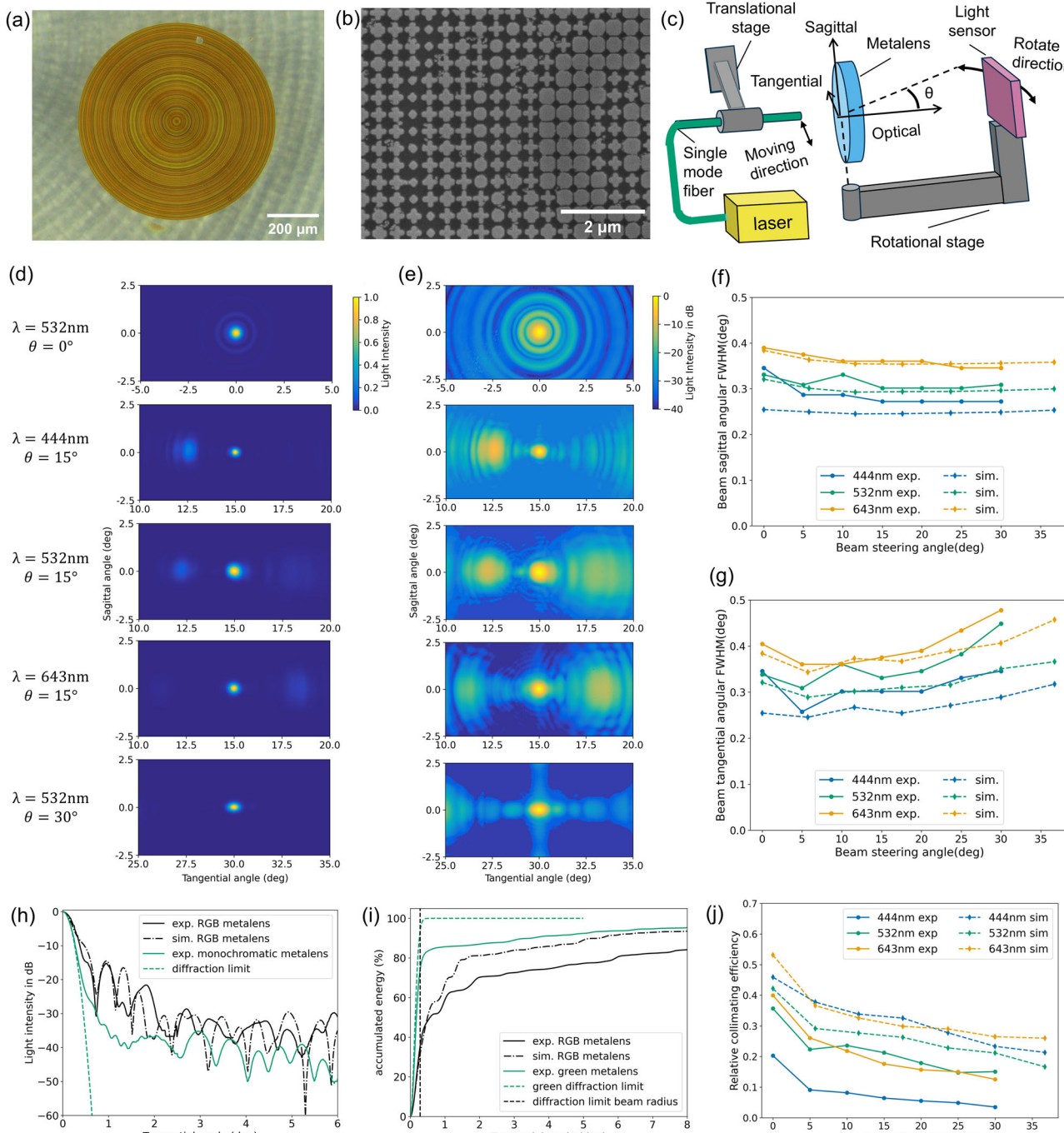

**Fig. 3 | Experimental optical characterization of the metalens. a** Top view optical microscope and **b** Top-down view SEM images of the fabricated RGB metalens. **c** Experimental measurement setup to determine the angular intensity distribution of the collimated beam at various steering angles. Experimental normalized angular beam intensity distribution for RGB wavelengths at steering angles $\theta = 0°$ and 15°. **d** is in linear scale while (**e**) is in log scale. The experiment, simulation, and diffraction-limited angular intensity FWHM of the steered beams in sagittal (**f**) and tangential (**g**) direction vs. the beam steer angle for 3 design wavelengths. **h** The one-dimensional cross-section intensity distribution vs. tangential angle. **i** Accumulated energy in percentage within a certain range of angle from the center of the beam. For (**h**) and (**i**), the steered beam is at green (532 nm) and is steered to $\theta = 0°$. The black solid (dashed) curve is the experiment (simulation) data from the RGB polychromatic metalens, while the blue solid (dashed) curve is the experiment (theoretical limit) data from a monochromatic metalens that has the same optical functionality but be designed to only work at green (532 nm). **j** Experimentally measured and simulated relative collimating efficiency vs. beam steering angle.

the beams are collimated and steered without chromatic dispersion to the same angle of 15°, matching simulations in Fig. 2. We note that for intermediary wavelengths the beams did not follow the same trend, as shown in the Supplementary Note 7 Fig. S8. This essentially verifies the intended polychromatic character of the metalens. To better illustrate the concurrent losses, we also plotted the angular beam intensity distribution on a log scale in Fig. 3e. Again, the measured angular intensity distributions match well to

the simulated ones in Fig. 2b, whereas losses primarily occur in the tangential direction with about a factor of 10 or much smaller than the intensity in the main lobe. We further calculated the experimental beam radii from the angular distributions (see Supplementary Note 8 Fig. S9) and presented the results in Fig. 3f, g. Across the required steering range, the experimental values are close for all three wavelengths to the simulation values and the diffraction limits, thus closely matching the desired functionality.

The performance of the polychromatic metalens is close to that of standard monochromatic metalens designed for 532 nm. To illustrate this, we directly compare the one-dimensional (1D) intensity cross-section along the tangential angular axis for on-axis collimation (i.e. $\theta = 0°$), as presented in Fig. 3h. It can be seen that at angles smaller than 0. 3° (the diffraction-limited steered beam's angular radius), the 1D angular intensity distribution of the polychromatic RGB metalens overlaps with the monochromatic metalens and the diffraction-limited distribution. However, at larger angles, the distribution of the RGB metalens drops slower than that of the monochromatic metalens and the diffraction limit. This indicates that the RGB metalens collimates the beam close to the diffraction limit, but distributes more energy outside of the FWHM of the beam, essentially leading to a lower efficiency. Figure 3i plots the accumulated energy vs. the angle by integrating the intensity vs. angle data in Fig. 3h. As can be seen, for the RGB metalens only around 40% of the energy is concentrated within the diffraction-limited angular radius. It is lower than the experimental relative efficiency of the monochromatic lens, which has an efficiency of around 80%.

We obtained the experimental relative collimating efficiency at $\theta = 0$ using the approach presented above. For $\theta \neq 0$, whose 2D angular intensity distributions are not central symmetric, the relative efficiency is calculated by measuring the absolute intensity of the steered beam at $\theta \neq 0$ and comparing with that of the steered beam at $\theta = 0$. This is valid, because the amount of light transmitted through the metalens does not vary with the steering angle $\theta$, which is supported by the simulation. Figure 3j shows the relative collimating efficiency of the RGB metalens vs. the beam steering angle. On average, our polychromatic metalens have an average relative efficiency of around 32% for on-axis (0° steering angle) among the 3 wavelengths, and an overall average efficiency of 13% across the entire 70° field-of-view (weighted by the solid angle). Specifically, red and green wavelengths show similar efficiencies around 40% for on-axis angle, which drops to around 15% for 30° steering angle. These efficiencies are only about 70–80% of the simulation results, which could be due to errors in fabrication and optimization model predicting the phase response and transmission of the meta-atoms, as well as inaccuracies in the beam propagation simulation based on the scalar-field diffraction theory. We note that unlike in the simulation results, the blue wavelength 444 nm has a notably lower experimental efficiency compared to red and green wavelengths, ranging from only 20% at 0° to 5% at 30°. This might be due to the fact that at this wavelength, the ratio of wavelength to the 300 nm pitch of the meta-atom is lower, resulting in a lower sampling rate of the light field by the metalens.

## RGB imaging test of the RGB metalens
We verified the imaging performance of our RGB metalens in an SFE testing platform, shown in Fig. 4a. Three different laser outputs are coupled into the same single-mode fiber, which is actuated by a piezo tube actuator. The beam emitted at the fiber tip is steered by the projecting lens to illuminate a checkerboard test pattern. The reflected light is collected by a detector to construct the image (Details in the method section). The RGB image of the pattern captured by the metalens is shown in Fig. 4b. We note that the image appears non-uniform due to the position of the sensor relative to the imaging target, and for an actual implementation of the SFE, several collection fibers are used simultaneously to form a uniform image. Therefore, we cropped the image to the upper right quadrant and displayed the respective color channels in the upper row of Fig. 4c. For comparison, we show images captured by using 3 different monochromatic metalenses designed for the corresponding wavelengths (lower row of the Fig. 4c). These images are linearly rescaled to enhance the visual representations. In Fig. 4d, we plot the greyscale value variation across the checkerboard pattern. The plot reveals that at the interface of the bright and dark squares, the relative gradient of pixel values for the RGB metalens closely matches that of the monochromatic lens, suggesting that the RGB metalens maintain a comparable imaging resolution. However, it is notable that for the RGB metalens, the contrast between the bright and dark blocks is less pronounced compared to

the monochromatic lens. We define the pattern contrast as:

$$\text{Contrast} = \frac{I_{\text{bright}} - I_{\text{dark}}}{I_{\text{bright}} + I_{\text{dark}}} \tag{3}$$

where $I_{\text{bright}}$ and $I_{\text{dark}}$ are the average pixel values of the bright and dark block of the checkboard pattern. The patterns captured by the RGB metalens have a low contrast by a factor of 4 for blue, 2.5 for green, and 2 times for red compared to that captured by the corresponding monochromatic metalens, owing to the low relative efficiency of the RGB metalens. The unprocessed images can be found in Supplementary Note 10 Fig. S11.

As can be seen in the images from the polychromatic RGB metalens, a blurry shadow of the checkboard patterns overlays with the clear pattern. This shadow defect is likely caused by spurious side lobes, as shown in Fig. 3e. These artifacts could be mitigated by image deconvolution. Specifically, the Richardson-Lucy deconvolution algorithm is a promising solution as it can accommodate for the position-dependent point spread function of our metalens. Alternatively, these artifacts could be reduced using a confocal SFE setup[35], in which the scanning fiber not only emits the light for illumination but also collects the back-scattered light that is re-focused by the metalens into the fiber. The confocal setup can spatially filter out the light not being collimated at the designed position, which greatly reduces the effect of the spurious side beams. Specifically, in conjunction with a double-clad fiber[36] with a single mode core for emission and a larger multimodal clad for collecting return light, the lower collection efficiencies of a confocal setup which would otherwise impede such a device could be mitigated.

## Conclusion
In this paper, we inverse designed polychromatic metalens for a scanning fiber endoscope, with a large 70° field-of-view (FOV) imaging capability, which has not been reported in the previously demonstrated polychromatic metalens. The beam steering metalens is suitable for an RGB SFE imaging system that collimates and steers the beam emitted from a scanning single mode fiber tip on the focal plane of the lens. We simulate and experimentally characterize the performance of the designed metalens, which shows no observable chromatic dispersion among the three design wavelengths 643 nm, 532 nm, and 444 nm. With an incident beam diameter of ~ 80 μm on the metalens (the effective aperture), the beam angular radius of the beams collimated and steered by the metalens are below 0. 5° across the three design wavelengths and a FOV of 60°, close to the diffraction limit, ensuring a high angular resolution of the SFE. While the elimination of the chromatic aberration compromises the relative collimating efficiency, which effectively lowers the imaging contrast, our metalens maintain a ~40% efficiency for red (643 nm) and green (532 nm) light and a ~20% efficiency for blue (444 nm) at 0° angle. These efficiency values are comparable to the previously reported polychromatic metalens. The comparison can be found in Supplementary Note 11. The efficiencies decrease with the beam steering angle and drop to ~15% for red and green and ~5% for blue at the boundary of the FOV. These compromised relative efficiencies result in a 2, 2.5, and 4 folds reduction in pattern contrast compared to a single wavelength metalens at red, green, and blue wavelengths. However, with improved fabrication technologies, these efficiencies could be improved by increasing the height of the meta-atoms or using a higher index material. Our work shows the feasibility of overcoming the chromatic aberration of the metalens in a large FOV endoscopic imaging system via an inverse design framework. These results help to promote the possible application of metalenses in endoscopes and other miniaturized imaging systems that benefit from the intrinsic advantages of metalenses, including light weight, low thickness, and ease of aperture reduction.

## Methods
### Ray optics simulation
The optical functionality of the beam steering metalens in our SFE system is obtained through ray optics simulation and optimization in

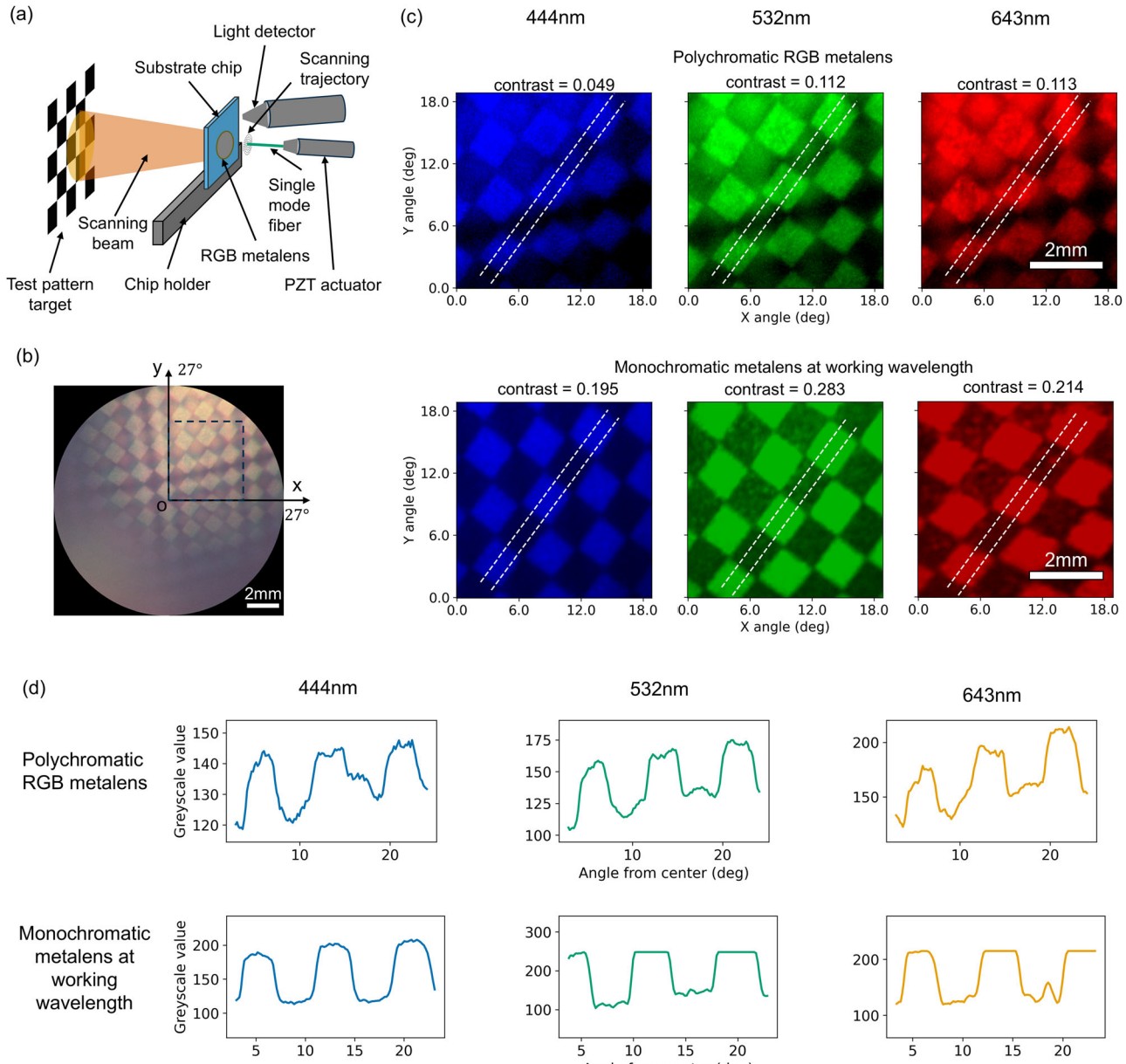

**Fig. 4 | Imaging test of the scanning fiber endoscope system using the metalens. a** Schematic demonstration of the scanning fiber endoscope imaging system. The imaging distance is 14.5 mm. The blocks of the checkerboard have a side length of 1.1 mm, and the total angular FOV of this SFE platform is 54°, limited by the actuation range of the single-mode fiber. **b** RGB tri-color image of the test pattern using our polychromatic metalens. **c** The upper row shows the blue, green, and red channel of the image in (**b**) within the zone circled by the dashed line square. The lower row plots the same pattern imaged by 3 monochromatic metalenses designed for the corresponding wavelengths. A linear mapping of the pixel values of the raw images to that of these displayed images is tailored to each specific image to enhance the visual representations. The pattern contrast is defined as the difference divided by the sum of the average pixel values of the bright and the dark blocks. **d** The greyscale value variations across the checkboard patterns along the two white dashed lines. The values are average over the region sandwiched by the two dashed lines.

Zemax. The metalens are modulated as a phase mask with an even polynomial phase profile. The beams emitted from the fiber tips are modulated as point sources with limited emitting angle defined by the NA of the fiber, which is set to be 0.18. We put 7 point sources evenly distributed along the tangential (y) axis from $y = 0\,\mu m$ to $y = 240\,\mu m$ on the object plane, which is $400\,\mu m$ away from the metalens. The phase profile of the metalens is optimized to minimize the root-minimum-square radius of the spot diagrams of the image of 7 point sources at far field (image distance is 80 mm). The spot diagrams of the steered beams corresponding to the 7 point sources can be found in Fig. S1. The corresponding positions of the steered beams on the metalens and the steering angles are summarized in Table S1.

**Inverse design implementation**

For the convenience of defining a loss function, we reverse the propagation directions of the steered beams in the SFE operation such that these beams become incident beams, as indicated in Fig. 1c. The reversion of the beam propagation can be justified as the system is reciprocal. We model the metalens as a phase and amplitude modulation mask whose local modulation of the light is determined by the local meta-atom. Starting from a configuration with random geometric parameters $a, b$ of the meta-atoms, we implement a fully-differentiable inverse design framework[30] to iteratively optimize the geometric parameters of all the meta-atoms across the entire metalens. This optimization aims to achieve the focusing of the light at R, G, and B wavelengths from different incident angles $\theta$ to the plane where the

fiber tip is physically scanned, as illustrated in Fig. 1. The angles and positions of the incident beams are set as the positions and beam steering angles of the steered beams in the SFE system, which are obtained from the ray-tracing simulation. This optimization is performed via using the gradient decent method to minimize a loss function, which is defined as:

$$L = -\sum_{\lambda,\theta} \log(\mathrm{SR}_{\lambda,\theta}) \tag{4}$$

where the summation is performed over seven beam steering angles $\theta$ and three wavelengths $\lambda$. Here $\mathrm{SR}_{\lambda,\theta}$ is the Strehl ratio of the incident beams at $\theta$ and $\lambda$. The Strehl ratio that defines the focusing efficiency of the lens is calculated as the maximal value of the simulated light intensity distribution $I_{\lambda,\theta}(x,y)$ on the focal plane divided by the peak intensity on the focal plane of an ideal lens. We empirically define this loss function because the percentage of the collimated incident light being focused on a spot is proportional to the Strehl ratio. In addition, we have tried different loss functions, and found that minimizing this negative summation of the log of the Strehl ratio will uniformly maximize the Strehl ratio at all incident angles and wavelengths. The intensity profile on the focal plane is calculated from the scalar field right after the beam transmits through the metalens via the angular spectrum method. This complex field can be calculated from the incident field and by modelling the metalens as a phase-mask:

$$A_{\mathrm{o}} = A_{\mathrm{i}} T_{\mathrm{m}} \tag{5}$$

$$\psi_{\mathrm{o}} = \psi_{\mathrm{i}} + \psi_{\mathrm{m}} \tag{6}$$

where $A_{\mathrm{o}}$ and $\psi_{\mathrm{o}}$ are the amplitude and phase of the field after passing through the metalens, $A_{\mathrm{i}}$ and $\psi_{\mathrm{i}}$ are the amplitude and phase of the incident field, $T_{\mathrm{m}}$ and $\psi_{\mathrm{m}}$ are the transmission and phase modulation to the light by the meta-atom at each lattice point. The intensity distributions on the optical (yz) plane for the final design can be seen in Supplementary Note 2 Fig. S2.

To perform the fully-differentiable inverse design framework to iteratively optimize the geometric parameters of all the meta-atoms across the entire metalens[30], a differentiable function of the meta-atom's phase retardations and amplitude modulations on the incident light with regard to its geometric parameters is essential. This function is approximated by a deep neural network (DNN) meta-model, because the DNN model has a better fitting capability to address drastic fluctuations of the phase and amplitude response of those meta-atoms that are in resonance modes. To train this DNN meta-model, 5673 training samples of the meta-atoms' phase and amplitude responses are calculated via finite-difference time-domain simulations with monochromatic plane wave incident and periodic boundary conditions. The height of the $Si_3N_4$-on-$SiO_2$ meta-atoms is 750 nm. The unit cell is square with a periodicity of 300 nm. The two tunable dimension parameters $a$ and $b$ range from 80 nm to 260 nm, and satisfy $80\ \mathrm{nm} \leq a \leq b \leq 260\ \mathrm{nm}$. The wavelengths are 643 nm, 532 nm, and 444 nm. The training dataset generated via finite-difference time-domain simulation is shown in Supplementary Note 3 Fig. S4. The details about the DNN meta-model training process can be found in Supplementary Note 4.

By building a DNN meta-model to map the geometries of meta-atoms to the phase and amplitude response, using the angular spectrum method to simulate the free-space propagation of the light and calculate the light intensity distribution on the focal plane, and defining a loss function to evaluate the performance of the metalens, we establish a mapping between the geometries of all the meta-atoms and the loss function. We then optimize the geometry of each individual meta-atom to minimize the loss function via a gradient decent-based method. More details about this inverse design framework can be found elsewhere[30].

## Fabrication of the metalens

The metalens are fabricated using an electron beam lithography process. A 750 nm thick $Si_3N_4$ is deposited on a 500 μm thick $SiO_2$ chip using plasma-enhanced chemical vapor deposition (SPTS Technologies Ltd., Delta LPX). An e-beam resist (ZEP-520A) is then spin-coated onto the chip at 5000 rpm and patterned by e-beam lithography (JEOL Ltd., JBX-6300FS). Subsequently, a ~85 nm thick $Al_2O_3$ hard mask is created by electron beam assisted evaporation (CHA Industries, SEC-600) followed by resist lift-off in N-methyl-2-pyrrolidone (NMP) solution at 90 °C overnight. Subsequently, the SiN layer is etched by a fluorine-based reactive ion process (Oxford, PlasmaLab 100, ICP-180).

## Measurements of angular intensity distributions of the steered beams

A single-mode fiber (Thorlabs, P1-460B-Fc-5) is mounted on a 3D translational stage to align with the metalens. The fiber is coupled to laser sources. A supercontinuum laser (NKT Photonics, SuperK Fianium FIR20, paired with SuperK SELECT 4× VIS/IR tunable filter) is used to provide monochromatic light at 485−643 nm, while a blue diode laser (Opto Engine LLC, MDL-III-445L 80 mW) is used to provide monochromatic light at 444 nm. A camera sensor (Allied Vision, Prosilica GT 1930C) is used to capture the image of the steered beams at 80 mm away from the metalens, measuring the angular intensity distribution of the beams at a far field. To measure the intensity across 5 orders of magnitude, we take multiple images of the same beam with different exposure times ranging from ~100 μs to 1 s. Subsequently, these images are processed such that the saturated pixels of the images taken with long exposure times are replaced with the unsaturated pixels of the images with shorter exposure times.

The 1D cross-section of the angular intensity distribution of the on-axis steered beam (steering angle $\theta = 0$) across 0° to 30° angle range is subtracted from the 2D angular intensity distribution across −2.5° to 2.5° sagittal angle range and −2.5° to 32.5° tangential angle range. Due to the limited size of our camera sensor, we can only measure a 5° × 10° angular field for a single exposure. Thus, to get an angular field range of 30°, we stitch 7 fields centered at tangential angles ranging from 0° to 30° with a step of 5°.

## SFE image test

The imaging performance of our metalens was tested on an SFE platform invented in our previous work[25] and built by VerAvanti Inc. In this platform, a single mode fiber is resonantly actuated by a homemade piezo tube (see Supplementary Note 9). The fiber tip position $(x, y)$ follows a spiral trajectory on the focal plane of the metalens given by:

$$x(t) = A_{\max}(t/T) \cdot \sin \omega t \tag{7a}$$

$$y(t) = A_{\max}(t/T) \cdot \cos \omega t \tag{7b}$$

Here $T$ is the period of one complete scan, $t \in [0, T]$, $\omega$ is the resonance angular frequency of the scanning fiber mechanical system, with $\omega T/(2\pi) \gg 1$, and $A_{\max}$ is the maximal lateral displacement of the fiber tip. The beam emitted from $(x, y)$ on the focal plane of the metalens is projected to $(x_I, y_I) = (\frac{d}{f}x, \frac{d}{f}y)$ on the image plane at the far field, where $d = 14.5$ mm is the imaging distance and $f = 0.4$ mm is the focal length. A light sensor detects the time-varying light intensity signal $I(t)$, which is synchronized with the beam scanning trajectory $x_I(t), y_I(t)$ to reconstruct the image signal $I(x_I, y_I)$. The beam scanning trajectory was calibrated prior to imaging.

## Data availability

The simulation and experimental characterization data of the designed polychromatic metalens that support the findings of this study are available in Zenodo with https://doi.org/10.5281/zenodo.14714084.

## Code availability

The codes used to generate the simulation and experiment results presented in this study are available in Zenodo with https://doi.org/10.5281/zenodo.14714084. The codes for the inverse-design optimization of the polychromatic metalens are available on GitHub: github.com/Luochenghuang/metabox.

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

## Acknowledgements

The work is supported by NSF GCR 2120774. Part of this work was conducted at the Washington Nanofabrication Facility / Molecular Analysis Facility, a National Nanotechnology Coordinated Infrastructure (NNCI) site at the University of Washington with partial support from the National Science Foundation (NSF) via awards NNCI-1542101 and NNCI-2025489. We thank VerAvanti Inc. for providing us with a testbed demonstrated in Fig. S11a to perform imaging tests.

## Author contributions

Ningzhi Xie did the overall designs and performed the numerical simulations and experimental characterizations of the metalens as well as the data processing. Zhihao Zhou performed the inverse-design optimization of the metalens and assisted in the numerical simulations of the metalens. Johannes Fröch and Matthew Carson assisted in building up the setup for experimental characterizations of the metalens. Arka Majumdar, Eric Seibel, and Karl Böhringer are the principal investigators who supervised the research and provided critical guidance throughout the project. Arka Majumdar is the corresponding author.

## Competing interests

The authors declare the following competing interests: Eric Seibel is the consultant to VerAvanti Inc., Redmond, WA, which has University of

Washington license for the SFE. Arka Majumdar and Karl F. Böhringer are co-founders of Tunoptix, which is looking into commercializing meta-optics. All other authors declare no competing interests.
