## [Transparent Peer Review file · Communications Engineering]

Large field-of-view polychromatic metalens for full-color scanning fiber endoscopy

Corresponding Author: Professor Arka Majumdar

Version 0:

Reviewer comments:

Reviewer #1

(Remarks to the Author)

In this work, the authors proposed an inverse-designed polychromatic metalens to address the challenges faced by conventional metalenses, particularly their strong chromatic aberrations, which limit their utility in multi-color imaging. The presented metalens, with a diameter of 680 μm , achieves low dispersion across three distinct wavelengths (643nm, 532nm, and 444nm), making it suitable for tri-color scanning fiber endoscopes. The metalens collimates and steers light emitted from a scanning fiber tip, providing illumination of red, green, and blue light, and supports tri-color imaging with a large field-of-view of 70°. The metalens achieves a close-to-diffraction-limited 0.5° angular resolution, restricted only by the effective aperture of the system. The average relative efficiency of the metalens for on-axis angles among the three design wavelengths is around 32%. This work holds promise for the application of metalenses in endoscopes and other miniaturized imaging systems. In my opinion, the technical content of this manuscript is relevant and the paper is well written. This paper brings new knowledge with detailed explanations which may be useful for the community. However, the following are my main concerns which need to be clarified before acceptance of this paper for publication:

1. The paper mentions that the designed metalens exhibits low dispersion across three distinct wavelengths. How is this low dispersion achieved? Did the authors employ design methods similar to conventional achromatic metalenses? Additionally, how does the chromatic aberration correction capability of the proposed polychromatic metalens compare to that of conventional achromatic metalenses? More detailed explanations are needed in this regard.
2. For the proposed polychromatic metalens, could the authors provide Z-axis beam profile scanning at different wavelengths (643 nm, 532 nm, and 444 nm)? This would help to visually demonstrate that the focal positions for the three distinct wavelengths are closely aligned, thereby providing stronger evidence of the metalens's low dispersion capability.
3. The inverse design method is mentioned briefly and is primarily based on a deep neural network (DNN) meta-model. Could the authors explain in more detail the differences between the training process and the optimization process? Additionally, regarding the loss function presented in Equation 4, could the authors elaborate on how this specific loss function was derived? A complete derivation should be included in the supplementary materials.
4. For the proposed DNN, how long does the entire deep learning training process take? During the testing process (predicting unseen data in training or validation process), approximately how much time is required? For each dataset, what are the input and ground truth data? The authors should provide more detailed explanations. Additionally, please specify the computer configuration used, including the CPU and GPU specifications, the deep learning framework (TensorFlow or PyTorch?), and the amount of RAM.
5. In this paper, the unit cell is designed in a cross shape. Why was this particular shape chosen? Why not use other types of unit cells, such as cylindrical or rectangular shapes? What are the advantages of the cross-shaped unit cell compared to other designs? The authors should provide more details on the rationale behind this choice.
6. How does the performance of this polychromatic metalens compare with existing technologies in terms of efficiency, angular resolution, and field-of-view? A comparative analysis with state-of-the-art refractive lenses and other types of metalenses would help contextualize the advancements presented in this paper.
7. The current number of references is insufficient. The authors are advised to supplement the paper with more references related to inverse design, achromatic metalens design, and the application of artificial intelligence in meta endoscopy, etc. Here are some references that need to be added to the article. Please include more references.
[1] Wang, Shuming, et al. "Broadband achromatic optical metasurface devices." *Nature Communications* 8.1 (2017): 187.
[2] Wang, Shuming, et al. "A broadband achromatic metalens in the visible." *Nature Nanotechnology* 13.3 (2018): 227-232.
[3] Fan, Zhi-Bin, et al. "A broadband achromatic metalens array for integral imaging in the visible." *Light: Science & Applications* 8.1 (2019): 67.
[4] Lin, Ren Jie, et al. "Achromatic metalens array for full-colour light-field imaging." *Nature Nanotechnology* 14.3 (2019): 227-231.
[5] Ren, Haoran, et al. "An achromatic metafiber for focusing and imaging across the entire telecommunication range."

Nature Communications 13.1 (2022): 4183.

[6] Luo, Yuan, et al. "Varifocal metalens for optical sectioning fluorescence microscopy." *Nano Letters* 21.12 (2021): 5133-5142.

[7] Luo, Yuan, et al. "Meta-lens light-sheet fluorescence microscopy for in vivo imaging." *Nanophotonics* 11.9 (2022): 1949-1959.

[8] Chu, Cheng Hung, et al. "Intelligent Phase Contrast Meta-Microscope System." *Nano Letters* 23.24 (2023): 11630-11637.

[9] Chia, Yu-Hsin, et al. "In Vivo Intelligent Fluorescence Endo-Microscopy by Varifocal Meta-Device and Deep Learning." *Advanced Science* (2024): 2307837.

8. What are the technical challenges or considerations that need to be addressed in order to precisely align the metalens with a Single Mode Fiber (SMF) in the practical setup of an SFE imaging system, given that the diameter of the proposed metalens is only 680 μm ? Additionally, could you provide detailed specifications for the SMF, such as its diameter?

9. Furthermore, regarding the piezo tube actuator (PZT), there is a lack of detailed explanation on the structure of the PZT, whether it is commercially sourced or homemade, neither in the main text nor in the supplementary materials. It is suggested that the authors provide a detailed schematic diagram illustrating the internal structure of the PZT.

10. In Figure 3 (a) of the metalens OM image, there appears to be a hole damaged. Would this hole damaged affect the overall efficiency and imaging quality of the metalens? Additionally, could the authors provide further explanation on the cause of this hole damaged?

11. The authors mention an average relative efficiency of around 32% for the on-axis angle. Could you provide further details on the methodology used to measure this efficiency and elaborate on the factors contributing to the losses? A comprehensive analysis of efficiency measurements, including identification of loss mechanisms and comparison with theoretical maximum efficiency, would enhance the depth of the efficiency claims. How does this efficiency compare to other state-of-the-art polychromatic metalenses?

12. The metalens is said to support a large field-of-view (FOV) of 70° with an angular resolution close to the diffraction limit of 0.5°. Can the authors provide experimental verification of these specifications? What methods were used to measure the FOV and angular resolution, and are there any trade-offs between these two parameters?

13. The internal structure of the DNN should be further elaborated upon, and it is suggested that the authors provide a detailed schematic diagram illustrating the model architecture.

14. How does the performance and computational efficiency of the inverse design approach compare to conventional design methods? What are the advantages that inverse design offers?

15. The clarity of Figure 4 (a) is insufficient, making it difficult to discern the placement of the metalens and the structure of the Single Mode Fiber (SMF). Could additional photographs from different angles be included, along with explanatory diagrams? Furthermore, the orientation of the light detector relative to the SMF appears to be oblique. Could the authors specify the angle of this obliqueness and elaborate on how this angle or setup is calculated?

16. The paper presents both simulation and experimental results for the metalens. Are there any significant discrepancies between the simulated and experimental data? If so, what might be the reasons for these discrepancies, and how can they be addressed in future work?

Reviewer #2

(Remarks to the Author)

Please refer to the attachment for details.

Version 1:

Reviewer comments:

Reviewer #1

(Remarks to the Author)

In this work, the authors present a novel design for polychromatic metalens optimized for tri-color scanning fiber endoscopy (SFE) applications. The authors employ an inverse design methodology to create a metalens that corrects chromatic aberrations across three distinct wavelengths (444 nm, 532 nm, and 643 nm). The metalens achieves efficient collimation and beam steering within a large field-of-view (FOV) of 70°, with an angular resolution close to the diffraction limit. The experimental results validate the theoretical model, demonstrating that the metalens can maintain high image quality with minimal chromatic dispersion across multiple wavelengths. This work holds promise for the application of metalenses in endoscopes and other miniaturized imaging systems. The manuscript is merit of publication. However, the following are my main concerns which need to be clarified before acceptance of this paper for publication:

1. Regarding the efficiency result of the metalens, the collimation efficiency falls below 5% at larger deflection angles. Under such conditions, can the imaging performance still be maintained at a satisfactory level?

2. In the final validation experiment about imaging checkboard patterns by metalens. The author mentions that deconvolution can be applied to remove blurry shadow effects caused by sidelobes. If the collimation efficiency at larger angles is significantly lower than at smaller angles, indicating larger sidelobes, is it still feasible to use deconvolution to effectively correct these artifacts in grayscale images under such conditions?

3. The meta-atoms are designed with a cross-shaped geometry, but in the SEM image shown in Fig. 3(b), this shape does not appear to match the intended design. The fabrication process is based on electron beam lithography (EBL). Could the disparity between the designed model and the fabricated structure of the meta-atoms be attributed to the aspect ratio between the structural parameters a and b, resulting in the four corner regions of the cross being too small to fabricate? The

paper mentioned that the cross-shaped meta-atoms were intended to help address chromatic aberrations, but how might the incomplete realization of this structure impact the accuracy of the experimental results?

4. For the proposed inverse design of the metalens, could the calculation of the loss function be explained in more detail? The paper notes that the contrast in the image is not discernible and even has the worst efficiency at the blue wavelength. Could this be due to insufficient phase compensation by the lens at this wavelength? Additionally, when using the deep neural network (DNN) to search for the optimal solution, how are the parameters that define the relationship between the RGB phase responses determined? Please clarify the distribution of the 5673 training samples, particularly the proportion of simulation results at different wavelengths, as the supplementary material suggests that the data distribution may not be uniform.

5. The paper mentions that the camera can only capture a $5^\circ \times 10^\circ$ angular range and multiple scenes must be stitched together to cover a 30° angular range. Could this stitching method introduce data inaccuracies or discontinuities that might affect the precision of the angular intensity distribution measurements?

6. In the experiment measuring the angular intensity distribution, the light sensor is rotated along a specified trajectory. Please explain how the sensor's rotation trajectory is calibrated to ensure that the light intensity signal is accurately captured at every angle.

7. The proposed polychromatic metalens has issues with efficiency and contrast compared to the monochromatic metalens. While you suggested using post-processing methods to enhance image quality, could you explain if there are possible adjustments in the design methodology that could make the polychromatic metalens more suitable for endoscopic imaging applications?

8. In Supplementary Figure 6, the polychromatic RGB metalens shows beam profiles for five different wavelengths. The beam profiles for 485 nm and 585 nm are larger, and noticeable ghosting occurs at an angle of 15° . Even at the design wavelength of 444 nm, ghosting is present. Could you explain the occurrence of these phenomena in the experimental results?

9. When the fiber illuminates along the spiral trajectory (x, y), according to the description in Section 4.5, if the fiber tip is adjusted by a piezo tube, the incident beam should form an angle and not simply shift parallelly up and down as in the simulation results. Could you provide an explanation for the differences between the light source conditions in the experiment and the simulation?

10. The paper mentions that the efficiency of the blue light is relatively lower, and the experimental results confirm this. However, in the simulation results (Fig. 2(f)), it is the green light that shows the lowest collimation efficiency. Moreover, in the experimental results, there is a dip at 5 degrees. Could you explain why this happens?

Reviewer #2

(Remarks to the Author)

This revised manuscript can be received.

However, the large field-of-view is one of the key innovations of this work, but there are few references related to large FOV metalens design. The authors should cite more works, like reference (1-3) and so on. And there is one detail that the authors should correct, that is, the shape of the complex transmission output should be $[2 \times 1]$ not $[3 \times 1]$ in Fig.S5a.

1 M. Y. Shalaginov, S. An, F. Yang, et al., "Single-Element Diffraction-Limited Fisheye Metalens." *Nano Letters* 20(10), 7429-7437 (2020).

2 S. L. Luo, F. Zhang, X. J. Lu, et al., "Single-layer metalens for achromatic focusing with wide field of view in the visible range." *Journal of Physics D: Applied Physics* 55(23), 235106 (2022).

3 Y. Liu, W.D. Li, K.Y. Xin, et al., "Ultra-wide FOV meta-camera with transformer-neural-network color imaging methodology" *Advanced Photonics* 6(5), 056001 (2024).

Version 2:

Reviewer comments:

Reviewer #1

(Remarks to the Author)

The current version can be recommended for acceptance.

Reviewer #2

(Remarks to the Author)

This revised manuscript can be received.

Reply to the Reviewers

Re: Manuscript ID COMMS-24-0224

“Large field-of-view polychromatic metalens for full-color scanning fiber endoscopy: Rebuttal letter for journal reviews”

First, we would like to thank the editor and reviewers for their time and effort to carefully reading our paper and providing us with constructive criticisms. Their comments have surely helped us improve the manuscript. Below you can find the changes made to the manuscript and our replies to all of the reviewers' remarks. We also have changed the manuscript accordingly.

Reviewer #1, comment #1

The paper mentions that the designed metalens exhibits low dispersion across three distinct wavelengths. How is this low dispersion achieved? Did the authors employ design methods similar to conventional achromatic metalenses? Additionally, how does the chromatic aberration correction capability of the proposed polychromatic metalens compare to that of conventional achromatic metalenses? More detailed explanations are needed in this regard.

Our response #1.1

We thank the reviewer for their questions, which help us improve the clarity of our paper. Here is our answer to these questions.

Our designed metalenses achieve low dispersion across three distinct wavelengths by performing a gradient based optimization to select meta-atoms from a large library of meta-atoms with diverse phase distributions to maximize the light being focused onto the focal plane for only these three wavelengths. This design method is not similar to a conventional achromatic metalens, in which a small set of meta-atoms with proper phase dispersion is selected to realize the required phase profiles that can focus the light at all wavelengths within the working wavelength range onto the focal plane (1, 2). We have revised the second and third paragraph of the introduction section of our manuscript to better explain the design approach of the polychromatic metalens and address the difference to the conventional achromatic metalens:

"To eliminate chromatic aberrations, the meta-atoms are often strategically engineered, ensuring that the phase imparted by these elements scales linearly with the wavelength. **With these meta-atoms, it is possible to realize the phase profiles that can simultaneously focus the light at all wavelengths within the working wavelength range onto the focal plane.**" "Polychromatic metalenses **with large diameters** have already been demonstrated by many groups **via selecting meta-atoms from a large library of meta-atoms with diverse phase distributions**"

Compared to the conventional achromatic metalens that has low dispersions for all wavelengths within the working wavelength range, our polychromatic metalens has low dispersions for only the three distinct designed wavelengths. The dispersions for the other wavelengths in between these wavelengths are large, as shown in Fig.S6, "Images directly captured by light sensor at various wavelengths and beam steering angles". However, this limitation of our metalens is purposeful since in the SFE the monochromatic laser light is used for illumination. We have revised the third paragraph of the introduction section to better explain this point:

"For a wide range of applications, light at multiple distinct wavelengths (e.g., from a laser) is used for imaging instead of broadband spectral light from the environment. In this case the imperative for a dispersionless metalens spanning a continuous wavelength band can be relaxed. A metalens ~~can be designed to satisfy~~ **that maintains** identical optical functionality for only multiple distinct wavelengths, **known as the polychromatic metalens, can satisfy the requirement of these applications.** Scanning fiber endoscopes (SFE) present such an application...."

Meanwhile, our metalens has a large diameter that contains hundreds of Fresnel zones, which is necessary for the wide field-of-view (FOV) of the SFE, but is not possible for the conventional achromatic metalens. This point has been addressed in the second paragraph of the introduction section: "However, this approach is limited to metalenses with an unwrapped phase profile (having only one Fresnel zone), where the product of the numerical aperture (NA) and the lens diameter falls within the same order of magnitude as the wavelength."

Reviewer #1, comment #2

For the proposed polychromatic metalens, could the authors provide Z-axis beam profile scanning at different wavelengths (643 nm, 532 nm, and 444 nm)? This would help to visually demonstrate that the focal positions for the three distinct wavelengths are closely aligned, thereby providing stronger evidence of the metalens's low dispersion capability.

Our response #1.2

We thank the reviewer for their suggestion on Z-axis beam profile scanning. However, we want to clarify that the primary function of our metalens is beam steering instead of focusing. This lens is only designed to deal with light incident within limited areas. For each different specific fiber position, this light incident area is different, and is much smaller than the aperture of the lens. Measuring the Z-axis beam profile with

planewave incidents on the whole metalens does not match with the designed figure of merit. We had already show in the manuscript that the lens simultaneously collimated the beam at 3 wavelengths and steered to the same angle. We believe this is sufficient to demonstrate the low dispersion of the metalens.

Reviewer #1, comment #3

The inverse design method is mentioned briefly and is primarily based on a deep neural network (DNN) meta-model. Could the authors explain in more detail the differences between the training process and the optimization process? Additionally, regarding the loss function presented in Equation 4, could the authors elaborate on how this specific loss function was derived? A complete derivation should be included in the supplementary materials.

Our response #1.3

We agree that the role of a DNN meta-model in the inverse design process was not explained clearly in our manuscript. We here clarify that, the DNN meta-model was not directly used in the inverse optimization process of the metalens. This inverse optimization process involves defining a loss function to evaluate the performance of the metalens, then perform a gradient based optimization on the geometric parameters of all the meta-atom to minimize the loss function. The role of the DNN meta-model was generating a differentiable function of the meta-atoms' phase retardations and amplitude modulations on the incident light with regard to their geometries, which is essential for the gradient based optimization. The DNN meta-model was first trained using the meta-atom library obtained via FDTD simulations. After this training, the DNN meta-model was set as a fixed, meta-atom geometries-to-phase(amplitude) mapping function in the inverse optimization process of the metalens. The differentiable meta-model is necessary because the FDTD simulated meta-atom library is discrete and thus non-differentiable while the gradient based optimization algorithm requires a differentiable function of the meta-atom's response to the incident light. We have made the following changes in the second paragraph of the section 4.2 to provide a better explanation:

~~"The transmission and phase response of the meta-atoms as a function of the dimension parameters a and b are calculated by a deep neural network (DNN) meta-model.~~ To perform the fully-differentiable inverse design framework to iteratively optimize the geometric parameters of all the meta-atoms across the entire metalens (3), a differentiable function of the meta-atom's phase retardations and amplitude modulations on the incident light with regard to its geometric parameters is essential. This function is approximated by a deep neural network (DNN) meta-model. To train this DNN meta-model, 5673 training samples of the meta-atoms' phase and amplitude responses are calculated via Finite-Difference Time-Domain (FDTD) simulations with monochromatic plane wave incident and periodic boundary conditions."

We mentioned in the last paragraph of section 4.2 that the Adam optimizer was employed for the training process (of the DNN meta-model), because the training was performed by minimizing the root-mean-square error between the DNN model predictions and the training data via gradient descent method (for which we use the Adam optimizer). We mention that the Adam optimizer was also utilized in the optimization process of the metalens, because this optimization is also performed via gradient descent method. However, we admitted that putting the detailed descriptions of the training of the DNN meta-model and the optimization of the metalens in the same paragraph is misleading. Therefore, we rewrote a paragraph about the details of the training of the meta-model and moved it to the supplementary section IV. In addition, we added a separate paragraph at the end of the section 4.2 to give an overall description of the inverse design of the metalens:

"By building a DNN meta-model to map the geometries of meta-atoms to the phase and amplitude response, using angular spectrum method to simulate the free-space propagation of the light and calculate the light intensity distribution on the focal plane, and defining a loss function to evaluate the performance of the metalens, we establish a mapping between the geometries of all the meta-atoms and the loss function. We then optimize the geometry of each individual meta-atom to minimize the loss function via a gradient decent based method. More details about this inverse design framework can be found elsewhere (3)."

Regarding the loss function for the gradient based optimization, it should quantitatively evaluate the performance of the metalens, which is the percentage of the fiber emitted light being collimated. We reverse the propagation direction of the light for the convenience of defining a loss function. Upon reversion of the light propagation, the percentage of the collimated light incident from different angles being focused to a spot on the focal plane should be maximized in the optimization. This percentage is proportional to the

Strehl ratio. Besides, we want uniform Strehl ratio across different wavelength and angles. Therefore, we empirically define the loss function as the negative summation of the log of the Strehl ratio at all incident angles and wavelengths. Minimizing this loss function will uniformly maximize the Strehl ratio at all incident angles and wavelength. We have revised the first paragraph of the section 4.2 to more clearly demonstrate the reason for choosing this loss function:

"The loss function is minimized when the metalens achieves a uniform Strehl ratio across different angles and wavelengths. We empirically define this loss function because the percentage of the collimated incident light being focused to a spot is proportional to the Strehl ratio. In addition, we have tried different loss functions, and found that minimizing this negative summation of the log of the Strehl ratio will uniformly maximize the Strehl ratio at all incident angles and wavelengths Strehl ratio."

Reviewer #1, comment #4

For the proposed DNN, how long does the entire deep learning training process take? During the testing process (predicting unseen data in training or validation process), approximately how much time is required? For each dataset, what are the input and ground truth data? The authors should provide more detailed explanations. Additionally, please specify the computer configuration used, including the CPU and GPU specifications, the deep learning framework (TensorFlow or PyTorch?), and the amount of RAM.

Our response #1.4

We agree with the reviewer that these details about the training process of the DNN should be included. We also note that we made a mistake about the number of layers of the DNN. It has 10 layers instead of 4. We have revised the last paragraph of the section 4.2 accordingly:

"The DNN architecture incorporates four layers of fully connected units, each layer consisting of 128 units with Rectified Linear Unit (ReLU) activation functions. The training process employs the Adam optimizer with a learning rate of 10^{-4} over 500 epochs. During the optimization process of the metalens, the Adam optimizer is utilized with a learning rate of 10^{-9} for a total of 50 epochs. The details about the DNN meta-model training process can be found in the Supplementary Information".

We add a new section in the supplementary information to include the details on the DNN model: The DNN architecture consists of a 10-layer fully connected network (FCN), with 128 units per layer and a Rectified Linear Unit (ReLU) activation function. The input features of the DNN comprised the incident wavelengths and the two geometric parameters of the meta-atom, while the output consisted of the real and imaginary components of the complex transmission. In the training process of the DNN meta-model, the dataset included the complex transmission (phase retardation and amplitude modulation) of 5,673 meta-atoms with different geometric parameters simulated using Lumerical FDTD, with 95% allocated for training and 5% for testing. The DNN networks were optimized by minimizing the mean squared error (MSE) between the predicted values and the ground truth. This optimization was performed using the Adam optimizer with a learning rate of 10^{-4} , and the model was implemented within the TensorFlow framework. The computational hardware consisted of a six-core CPU operating at 2.40 GHz, 23 Gigabytes of RAM, and one NVIDIA Tesla V100 GPU. The training process required 13 minutes for 500 iterations, while prediction generation during validation took 27 milliseconds."

Reviewer #1, comment #5

In this paper, the unit cell is designed in a cross shape. Why was this particular shape chosen? Why not use other types of unit cells, such as cylindrical or rectangular shapes? What are the advantages of the cross-shaped unit cell compared to other designs? The authors should provide more details on the rationale behind this choice.

Our response #1.5

We agree that the rationale behind the choice of the cross shape for the meta-atom could be improved, and thank the reviewer for pointing this out. We have revised the fifth paragraph of the section 2.1 accordingly:

"This meta-atom shape effectively diversifies the attainable phase range for the three wavelengths while still being simple enough to be reliably fabricated, as indicated by previous works. Compared to simple square or cylindrical shapes, this cross shape meta-atom diversifies the attainable phase range for the three wavelengths as indicated by previous works (3). Compared to other more complicated shapes, the cross shape is still simple enough to be reliably fabricated."

Reviewer #1, comment #6

How does the performance of this polychromatic metalens compare with existing technologies in terms of efficiency, angular resolution, and field-of-view? A comparative analysis with state-of-the-art refractive lenses and other types of metalenses would help contextualize the advancements presented in this paper.

Our response #1.6

The polychromatic metalens in this manuscript has a similar close-to-diffraction limited angular resolution and field-of-view compared to the state-of-the-art refractive lens assembly used for scanning fiber endoscope (SFE), but lower efficiency of $\sim 33\%$ compared to close to 100% of that of the refractive lens assembly. However, the metalens has a much thinner thickness, and its diameter can be easily scaled down. These unique advantages make it a promising substitution of the bulkier refractive lens (4), as we discussed in the first paragraph of the introduction section: "The diameter of a metalens can be arbitrarily scaled down due to its fabrication through two-dimensional lithography. These distinctive attributes position the metalens as a promising candidate for supplanting traditional refractive lenses in optical imaging systems, presenting a compelling avenue for system miniaturization."

We summarize the performance of our polychromatic metalens in comparison with other previously reported polychromatic metalenses in the following table. We have included this table also in the supplementary materials of the paper.

Effective aperture (μm)	Angular resolution (Experiment)	Angular resolution (Diffr. limit)	Angular FOV (Experiment)	Number of wavelengths	On-axis average efficiency	Source
300	$0.27^\circ @ 915\text{nm}$	0.18°	Near axis	2	$\sim 43\%$	Optica 2016(5)
600	$0.16^\circ @ 1300\text{nm}$	0.13°		3	$\sim 11\%$	Sci. 2015(6)
263	$0.37^\circ @ 915\text{nm}$	0.21°		3	$\sim 34\%$	Sci. Rep. 2016(7)
400	$0.12^\circ @ 690\text{nm}$	0.10°		3	$\sim 33\%$	Nano. L. 2018(8)
2000	$0.028^\circ @ 658\text{nm}$	0.019°		3	$\sim 15\%$	Nat.Comm.2022(9)
80	$0.41^\circ @ 643\text{nm}$	0.34°	$\sim 60^\circ$	3	$\sim 32\%$	This work

Table 1: Summary of the performance of various polychromatic metalenses

For the previously reported polychromatic metalenses, they all acted as focusing lenses, where the collimated light was incident on the metalenses, and the point spread functions (PSFs) at the focal plane were measured. We calculate experimental angular resolutions of these lenses as d_{FWHM}/f , where d_{FWHM} is the full-width-at-half-maximum (FWHM) diameter of the PSF. For our polychromatic metalens, it primarily acted as a collimation lens, where the light was emitted from a point source at the focal plane, and the PSF was measured at the far field. The experimental angular resolution is defined as θ_{FWHM} , which is the angular FWHM of the PSF at the far field. The diffraction limited angular resolution for a lens is $1.029\lambda/D_{\text{eff}}$, where λ is the wavelength and D_{eff} is the effective aperture. Our polychromatic metalens has a larger absolute angular resolution, but is still close to the diffraction limit, similar to other previously reported polychromatic metalenses. Note that for our metalenses, the effective aperture is not the same as the diameter of the metalenses, as only a small portion of the metalens is illuminated by the incident light. This results in a smaller effective aperture and thus a lower angular resolution, but also leads to the large angular FOV that unique to the other polychromatic metalenses. This is because the spatial separation of the lens regions that collimate light at different angles effectively reduces the off-axis aberrations. Our metalens has a comparable on-axis average efficiency to other polychromatic metalenses.

We revised the last paragraph of the introduction section to address that our metalens has a large FOV compared to other polychromatic metalenses presented in the papers we cited:

"Polychromatic metalenses have already been demonstrated by many groups. An inverse design framework has been employed to create polychromatic metalenses for thermal imaging, coherent fiber bundle endoscopy, and augmented reality display. However, these polychromatic metalenses only work for near-axis imaging. In this paper, we report an inverse designed ~~large field of view~~ (FOV $\sim 70^\circ$) polychromatic metalens as a beam steering lens that can support \$\sim 70^\circ\$ field of view (FOV) imaging in a RGB-SFE system."

We also have revised the conclusion section to address our advancement, added this comparison table and discussion to the supplementary information:

"In this paper, we inverse designed a polychromatic metalens for a scanning fiber endoscope, with a large 70° field-of-view ~~steering range~~. (FOV) imaging capability, which has not been reported in the previously demonstrated polychromatic metalens."

"While the elimination of the chromatic aberration compromises the relative collimating efficiency, which effectively lowers the imaging contrast, our metalens maintains a ~ 40% efficiency for red (643 nm) and green (532 nm) light and a ~ 20% efficiency for blue (444 nm) at 0° angle. These efficiency values are comparable to the previously reported polychromatic metalens. The comparison can be found in the supplementary information section XI. The efficiencies decrease with the beam steering angle and drops to ~ 15% for red and green and ~ 5% for blue at the boundary of the FOV..."

Reviewer #1, comment #7

The current number of references is insufficient. The authors are advised to supplement the paper with more references related to inverse design, achromatic metalens design, and the application of artificial intelligence in meta endoscopy, etc. Here are some references that need to be added to the article. Please include more references.

Our response #1.7

We agree that more relevant papers should be cited. We have already cited some of the papers recommended by the reviewer, and have added more citations in the introduction section:

"To eliminate chromatic aberrations, the meta-atoms are often strategically engineered, ensuring that the phase imparted by these elements scales linearly with the wavelength ~~(1, 2)~~ (1, 2, 10, 11, 12)." "Leveraging their ultra-thin thickness and the ease of aperture reduction, metalenses have been integrated into various kinds of endoscopic systems, including coherent fiber bundle endoscopes (13, 14), fluorescence confocal endoscopes ~~(15, 16)~~(15, 16, 17, 18), optical coherence tomographic endoscopes (19), and scanning fiber endoscopes (20)"

Reviewer #1, comment #8

What are the technical challenges or considerations that need to be addressed in order to precisely align the metalens with a Single Mode Fiber (SMF) in the practical setup of an SFE imaging system, given that the diameter of the proposed metalens is only 680 μm? Additionally, could you provide detailed specifications for the SMF, such as its diameter?

Our response #1.8

We agree that the alignment of the fiber with the metalens is non-trivial and need detailed elaboration. To align the single mode fiber (SMF) with the metalens, the fiber was mounted on a three-dimensional translational stage. For the lateral alignment, an objective lens, a tube lens, and a camera sensor were used to image the metalens. The fiber was moved laterally until the metalens image demonstrated that the center of the metalens was illuminated by the light emitted from the fiber. For the axis alignment, the camera sensor was put directly behind the metalens (without any lens in between), then the fiber was moved along the optical axis until the beam size on the camera sensor reached minimum. This indicated that the beam after transmitting through the metalens was collimated, so that the fiber tip was right at the focal plane of the metalens. We had added this into the supplementary information. The SMF is an off-the-shelf product (Thorlabs, P1-460B-Fc-5), which has a mode field diameter of 4.1 μm at 488 nm.

Reviewer #1, comment #9

Furthermore, regarding the piezo tube actuator (PZT), there is a lack of detailed explanation on the structure of the PZT, whether it is commercially sourced or homemade, neither in the main text nor in the supplementary materials. It is suggested that the authors provide a detailed schematic diagram illustrating the internal structure of the PZT.

Our response #1.9

We agree that more detailed information of the piezo tube actuator needs to be included. This actuator is homemade, and has already been used for many years. Thus it is not a new contribution for this paper. The schematic diagram of the actuator has been added into the supplementary information, along with a description on the fabrication of the piezo actuator:

"The base and the fiber mount are machined then glued to the piezo tube. Then the wires are soldered to the gold-plating on the piezo tube. Finally, the fiber is threaded through and glued into place with epoxy. More detailed characterization of the piezo tube can be found elsewhere (21)."

Reviewer #1, comment #10

In Figure 3 (a) of the metalens OM image, there appears to be a hole damaged. Would this hole damaged affect the overall efficiency and imaging quality of the metalens? Additionally, could the authors provide further explanation on the cause of this hole damaged?

Our response #1.10

The effect of the hole damage on the overall efficiency and imaging quality of the metalens should be pretty small. Because this hole only has a diameter of 20 μm , while the diameter of the area being illuminated by the light emitted from the fiber is $\sim 80 \mu\text{m}$, and the whole metalens has a diameter of 680 μm . This means the image within only about 10° out of the 60° FOV will be affected. And as the hole only occupies 1/16 of the area being illuminated, we expected the efficiency only has a fluctuation of about 1/16 of the original value. The cause of this hole damaged is probably a dust particle accidentally adhered onto the metalens pattern area when the EBL pattern was transferred onto the hardmask via a lift-off process. This dust particle prevented the hardmask material from adhering to the substrate via evaporation. Therefore, in the following lift-off process, the hardmask pattern was not formed in the area where the dust particle settled.

Reviewer #1, comment #11

The authors mention an average relative efficiency of around 32% for the on-axis angle. Could you provide further details on the methodology used to measure this efficiency and elaborate on the factors contributing to the losses? A comprehensive analysis of efficiency measurements, including identification of loss mechanisms and comparison with theoretical maximum efficiency, would enhance the depth of the efficiency claims. How does this efficiency compare to other state-of-the-art polychromatic metalenses?

Our response #1.11

For measuring the efficiency of the on-axis beam, we first measured the radial 1D intensity distribution vs. the angle at the far field, which was partially shown in Fig.3h. Subsequently, we integrated this intensity distribution, from 0 deg to 30deg, to get the curve of the accumulated energy that fall within the certain angle, as demonstrated in Fig.3i. We then found the percentage of the energy falling within the FWHM radius of the beam spot, which was defined as the efficiency. For measuring the efficiency of the off-axis beam (steering angle larger than 0), the efficiency was calculated by measuring the absolute intensity of the steered beam at that specific angle and comparing it with that of the on-axis beam. This methodology of efficiency measurement has been described in detail in the fourth and fifth paragraph of the section 2.3 "Characterization of the metalens" in the manuscript.

We note that the polychromatic metalens scatters a large portion of the energy outside of the FWHM of the beam, essentially leading to a lower efficiency. This could be due to the fact that the phase retardation and transmission of the meta-atoms does not fully match the required phase profile of an ideal lens under the restriction of the polychromatic requirement. Because of this, a certain portion of the light will inevitably be scattered to other regions outside of the collimated beam spot. We currently have no mean to determine a theoretical maximal portion of light that can be focused under the restriction of a polychromatic metalens. Compared to other polychromatic metalenses we cited in the manuscript, the efficiency values are generally close, which falls in the 15% – 40%*region*, as summarized in table 1.

Reviewer #1, comment #12

The metalens is said to support a large field-of-view (FOV) of 70° with an angular resolution close to the diffraction limit of 0.5° . Can the authors provide experimental verification of these specifications? What methods were used to measure the FOV and angular resolution, and are there any trade-offs between these two parameters?

Our response #1.12

We have already experimentally demonstrated the angular resolution and FOV in the manuscript. We measured the angular PSF at far field from 0° to 30° , using the setup in Fig. 3c, and plot the beam FWHM

vs angle in Fig.3f-g. This verifies the angular resolution of 0.5° and a FOV of at least 60° , slightly smaller than the designed FOV of 70° . We did not perform the experimental measurement up to the design 70° FOV due to the limited rotational angle of the rotational stage. We also performed RGB SFE imaging using the metalens we fabricated, as demonstrated in Fig.4b. By measuring the actual physical size of the checkboard pattern blocks, counting the number of the blocks in the image, and measuring the distance from the pattern to the metalens, we found the image has a FOV of 54° .

As for the trade-off between the FOV and the angular resolution, as we discussed in the response of the 6th comment, the limited diameter of the effective aperture (the area being illuminated by the light emitted from the fiber) allows for a wide FOV operation, but leads to lower angular resolution.

Reviewer #1, comment #13

The internal structure of the DNN should be further elaborated upon, and it is suggested that the authors provide a detailed schematic diagram illustrating the model architecture.

Our response #1.13

We agree that a detailed schematic diagram of the DNN architecture should be added. We have now added such a diagram in the supplement information section IV.

Reviewer #1, comment #14

How does the performance and computational efficiency of the inverse design approach compare to conventional design methods? What are the advantages that inverse design offers?

Our response #1.14

For conventional forward design, most of the computational resource is used for building the meta-atom library via Lumerical; the optimization of the meta-lens by locally minimizing the phase error of each meta-atom is fairly quick. For our inverse design, additional computational resources are needed for calculating the free-space propagation of the light and iteratively optimizing the meta-atoms via a gradient based method. These additional computational resources scale with the area of the metalens. Although the inverse design requires more computational resources, it offers many unique advantages. First of all, it globally optimizes the meta-lens, which will more likely lead to a better design compared to the local optimization in conventional forward design. Secondly, via tuning the loss function, inverse design can have a direct and more flexible control on the performance of the meta-lens. For example, in our case, we achieved more uniform lens efficiency across different wavelengths and incident angles using inverse design, while this kind of control is not possible when applying forward design that simply minimizes the phase error of each meta-atom. What's more, inverse design can deal with the situation when forward design is difficult as the phase profile of the lens cannot be obtained via ray-tracing or any other approach, although this situation does not apply to the research in this paper. However, the discussion of the advantages of the inverse design compared to the forward design is not the central topic of this manuscript. More details about the advantages of the inverse design are addressed in our group's other papers:(3).

Reviewer #1, comment #15

The clarity of Figure 4 (a) is insufficient, making it difficult to discern the placement of the metalens and the structure of the Single Mode Fiber (SMF). Could additional photographs from different angles be included, along with explanatory diagrams? Furthermore, the orientation of the light detector relative to the SMF appears to be oblique. Could the authors specify the angle of this obliqueness and elaborate on how this angle or setup is calculated?

Our response #1.15

We agree with the reviewer that the Fig.4a does not well illustrate the experimental setup of the SFE system. However, the experiment was performed on the setup built by the startup company VerAvanti, which does not want us to use photographs of their setup. Therefore, we replaced Fig.4a with a schematic diagram.

The orientation of the light detector is oblique just to avoid the collision of the detector with the translational stage (not shown in the figure, for alignment of the fiber and the metalens) that holds the piezo

actuator, while will still be able to put the detector close enough to the fiber to emulate the returning fiber in a real SFE device illustrated in Fig.1a.

Reviewer #1, comment #16

The paper presents both simulation and experimental results for the metalens. Are there any significant discrepancies between the simulated and experimental data? If so, what might be the reasons for these discrepancies, and how can they be addressed in future work?

Our response #1.16

We notice that the experimental efficiencies are overall lower than the simulation results. We attributed this to the fabrication imperfections as well as the inaccuracies of the beam propagation simulation based on the scalar-field diffraction theory, as we already discussed in our manuscript. Fabrication imperfections, such as the rounded shape of the corners of the meta-atom, and the random deviation of the fabricated meta-atom geometries from the design values, could be characterized and taken into account in the simulation to reduce the discrepancies between the simulation and experimental results. We also note that the experimental efficiency at 444 nm is more significantly lower than the simulation compared to the other two wavelengths. We had a hypothesis on this: At this wavelength, the ratio of wavelength to the 300 nm pitch of the meta-atom is lower. As we use this pitch as the spatial sampling rate of the complex field in the simulation of the beam propagation, such a short wavelength might result in insufficient spatial sampling rate that causes a significant discrepancy between the simulation and the experiment. This problem might be addressed by performing FDTD simulation to study the phase and amplitude of the light in between the meta-atoms and then building a meta-model to approximate this phase and amplitude as a function of the several nearby meta-atoms. With this meta-model, we can increase the sampling point in the simulation of the beam propagation using the angular spectrum method.

Reviewer #2, comment #1

According to my understanding, the design approach of the RGB polychromatic metalens for the SFE is as follows: firstly, the initial phase profile at each wavelength of RGB is obtained by using Zemax commercial software with even polynomial, then the initial structure arrangement aiming at minimizing the phase error is screened out based on the SiN cross-shape meta-atom library, and finally the geometric parameters of all the meta-atoms across the entire metalens is optimized by using the DNN meta-model and a gradient decent method to maximize the collimated output for the desired angular range. However, think more carefully, the whole design scheme can be achieved by just using the simplest traversal method without a neural network to model and proxy such a single-structure two-parameter meta-atom library. So, what are the advantages of the DNN meta-model here? Is it possible to speed up the optimization of structures, especially when the metalens size is further increased to mm or even cm level? If I understand correctly, this fully-differentiable architecture does not seem to make the most of the DNN's inverse design capabilities, such as directly predicting the one-to-many generation of structures according to the target complex amplitude. The term "inverse design" is not prominent enough in this paper, maybe it is easy to misunderstand that the design process is more inclined to a forward optimization loop. Also, in the manuscript, there is no comparison between this work and a traditional scanning fiber endoscope in terms of size or imaging performance. About the above, it is recommended that the authors add the relevant discussion to enhance innovation and persuasiveness.

Our response #2.1

We thank the reviewer for their careful reading and critical comments of our manuscript. We here response to these comments point by point.

For the DNN meta-model, it is built to make the mapping between the geometries of the meta-atoms and its complex response differentiable. While we agree that such a mapping function in principle could be approximated via a more simpler method such as fitting a polynomial proxy functions, we want to address that for a complex cross-shape meta-atom library with two individual variables, there is an increasing number of resonant meta-atoms, which might result in strong fluctuations of the phase and amplitude responses with respect to the geometric parameters of the meta-atom. A DNN has a better fitting ability on these complex two-variables functions. We revised the second paragraph of the section 4.2 to better justify the use of the DNN model:

~~"The transmission and phase response of the meta-atoms as a function of the dimension parameters a and b are calculated by a deep neural network (DNN) meta-model. To perform the fully-differentiable inverse design framework to iteratively optimize the geometric parameters of all the meta-atoms across the entire metalens (3), it is essential to acquire differentiable functions of the meta-atoms' phase retardations and amplitude modulations on the incident light with regard to its geometric parameters. These functions are approximated by a deep neural network (DNN) meta-model, because the DNN model has a better fitting capability to address drastic fluctuations of the phase and amplitude response of those meta-atoms that are in resonance modes."~~

To speed up the optimization process, our group is working on reducing the size of the optimization process by leveraging the rotational symmetry of the lens design, such as the Hankel transform.

Our design process is not a forward optimization, because we do not optimize the geometry of the meta-atoms according to the target complex amplitude of the metalens, as this target complex amplitude cannot be completely fulfilled due to the insufficient phase diversity of the meta-atom. A forward optimization typically directly minimizes the phase error to the target complex amplitude. However, the minimal local phase error at every site of the meta-atom does not necessary lead to the overall good performance of the metalens, as the light intensity distribution on the focal plane is cohesively affected by the phase and amplitude of all the meta-atoms. Therefore, we implement an "inverse design", where the light intensity distribution on the focal plane is calculated, and a loss function is defined from this light intensity distribution to evaluate the performance of the metalens. This loss function is then minimized in the process of iteratively optimizing the geometry of all meta-atoms, which is a typical "inverse design" process. We revised the last paragraph of the section 2.1 to better justify to use of inverse design instead of forward design:

~~"Using these meta-atoms, we first choose a meta-atom at every local site of the metalens by minimizing minimize the phase error between the desired and the achievable phase distribution for all three wavelengths. However, this local forward optimization does not necessary lead to the overall good performance of the~~

metalens, as the light intensity distribution on the focal plane is cohesively affected by the phase and amplitude of all the meta-atoms. Therefore, we implement an inverse design framework, This design is then used as the initial condition in our inverse design framework, as shown in Fig. 1c, to concurrently optimize the geometric parameters of all meta-atoms at the desired wavelengths. with the objective to maximize the collimated output for the desired angular range.— The forward optimization design is used as the initial condition in the inverse design. The inverse design objective is to maximize the collimated output for the desired angular range. This optimization is performed using a gradient decent method to minimize a loss function that evaluates the performance of the metalens. Further details on the inverse design method are described in the section 4.2."

When compared with a traditional scanning fiber endoscope (SFE) using a refractive lens, using a metalens reduces the length of the beam steering system from 1.2 mm to 0.86 mm, as demonstrated in our previous work (20). Of the 0.86 mm, 0.46 mm is attributed to the thickness of the the substrate that holds the metalens. Therefore, this length could be further reduced to 0.45 mm if the thickness of the substrate be further cut down to about 50 μm . We are working on this by trying to embed the metalens in a thin, flexible polymer material. Our results in this manuscript show that the RGB SFE using a polychromatic lens has the same FOV and diffraction limited resolution compared to the traditional refractive lens based SFE, while the imaging contrast is reduced due to the lower efficiency of the metalens compared to the refractive lens.

Reviewer #2, comment #2

In Section 2.2, Fig. 2a-c showed simulated beam intensity distributions in the y-z plane for beam steer angles only from 0° to 17.5° , but the maximum designed beam steer angle is 35° . Similar problems exist in the experimental results of Fig. 3d in Section 2.3. There are some inconsistencies and the results in the y-z plane for maximum beam steer angles are recommended to add.

Our response #2.2

We added the simulated and experimental beam intensity distribution in yz and transverse plane at the maximal beam steer angles at 532 nm to the Fig.2 and Fig.3 to fix this confusion.

Reviewer #2, comment #3

What is the accuracy of the DNN meta-model used in this work? Can it be demonstrated in the missing Fig. S4 (c) and (d) in Supplementary Information Section III?

Our response #2.3

We agree that the accuracy of the DNN meta-model should be demonstrated by plotting the DNN model predicted phase and amplitude response of the meta-atom vs the FDTD library. We have included this plot in the supplementary information section IV.

Reviewer #2, comment #4

If possible, in Supplementary Information Section V, the authors can consider adding corresponding phase response distributions like Fig. S4 (a), which will help the readers to understand the impact of the meta-atoms height and material index on the phase diversity more intuitively.

Our response #2.4

We agree that it is a good idea to plot the phase response distributions of the meta-atom with different height and material index to visualize the phase diversity. We have added this plot in the supplementary information.

References

- [1] W. T. Chen, A. Y. Zhu, V. Sanjeev, M. Khorasaninejad, Z. J. Shi, E. Lee, and F. Capasso. A broadband achromatic metalens for focusing and imaging in the visible. *Nature Nanotechnology*, 13(3):220–226, 2018. ISSN 1748-3387. doi: 10.1038/s41565-017-0034-6. URL <GotoISI>://WOS:000427009000015.
- [2] S. M. Wang, P. C. Wu, V. C. Su, Y. C. Lai, M. K. Chen, H. Y. Kuo, B. H. Chen, Y. H. Chen, T. T. Huang, J. H. Wang, R. M. Lin, C. H. Kuan, T. Li, Z. L. Wang, S. N. Zhu, and D. P. Tsai. A broadband achromatic metalens in the visible. *Nature Nanotechnology*, 13(3):227–232, 2018. ISSN 1748-3387. doi: 10.1038/s41565-017-0052-4. URL <GotoISI>://WOS:000427009000016.
- [3] Luocheng Huang, Zheyi Han, Anna Wirth-Singh, Vishwanath Saragadam, Saswata Mukherjee, Johannes E Fröch, Quentin A A Tanguy, Joshua Rollag, Ricky Gibson, Joshua R Hendrickson, Philip W C Hon, Orrin Kigner, Zachary Coppens, Karl F Böhringer, Ashok Veeraraghavan, and Arka Majumdar. Broadband thermal imaging using meta-optics. *Nat. Commun.*, 15(1):1662, February 2024.
- [4] Sarosh Irfan Madhani, Yuan Ping Chang, Catherine Olivo, Yang Liu, Eric J. Seibel, and Luis E. Savastano. Demonstration of navigational performance of preclinical angioscope designs in neurovascular phantoms. In *Biomedical Engineering Society Annual Meeting, Cardiovascular Engineering, Poster E-169*, Seattle, WA., October 2023.
- [5] E. Arbabi, A. Arbabi, S. M. Kamali, Y. Horie, and A. Faraon. Multiwavelength polarization-insensitive lenses based on dielectric metasurfaces with meta-molecules. *Optica*, 3(6):628–633, 2016. ISSN 2334-2536. doi: 10.1364/Optica.3.000628. URL <GotoISI>://WOS:000378847400012.
- [6] F. Aieta, M. A. Kats, P. Genevet, and F. Capasso. Multiwavelength achromatic metasurfaces by dispersive phase compensation. *Science*, 347(6228):1342–1345, 2015. ISSN 0036-8075. doi: 10.1126/science.aaa2494. URL <GotoISI>://WOS:000351219600036.
- [7] E. Arbabi, A. Arbabi, S. M. Kamali, Y. Horie, and A. Faraon. Multiwavelength metasurfaces through spatial multiplexing. *Scientific Reports*, 6, 2016. ISSN 2045-2322. doi: ARTN3280310.1038/srep32803. URL [HYPERLINK"http://gateway.isiknowledge.com/gateway/Gateway.cgi?GWVersion=2&SrcAuth=ResearchSoft&SrcApp=EndNote&DestLinkType=FullRecord&DestApp=WOS&KeyUT=000382492900001"](http://gateway.isiknowledge.com/gateway/Gateway.cgi?GWVersion=2&SrcAuth=ResearchSoft&SrcApp=EndNote&DestLinkType=FullRecord&DestApp=WOS&KeyUT=000382492900001)<GotoISI>://WOS:000382492900001.
- [8] Z. J. Shi, M. Khorasaninejad, Y. W. Huang, C. Roques-Carmes, A. Y. Zhu, W. T. Chen, V. Sanjeev, Z. W. Ding, M. Tamagnone, K. Chaudhary, R. C. Devlin, C. W. Qiu, and F. Capasso. Single-layer metasurface with controllable multiwavelength functions. *Nano Letters*, 18(4):2420–2427, 2018. ISSN 1530-6984. doi: 10.1021/acs.nanolett.7b05458. URL <GotoISI>://WOS:000430155900032.
- [9] Z. Y. Li, R. Pestourie, J. S. Park, Y. W. Huang, S. G. Johnson, and F. Capasso. Inverse design enables large-scale high-performance meta-optics reshaping virtual reality. *Nature Communications*, 13(1), 2022. doi: ARTN240910.1038/s41467-022-29973-3. URL <GotoISI>://WOS:000790385800005.
- [10] Zhi-Bin Fan, Hao-Yang Qiu, Han-Le Zhang, Xiao-Ning Pang, Li-Dan Zhou, Lin Liu, Hui Ren, Qiong-Hua Wang, and Jian-Wen Dong. A broadband achromatic metalens array for integral imaging in the visible. *Light Sci. Appl.*, 8(1):67, July 2019.
- [11] Ren Jie Lin, Vin-Cent Su, Shuming Wang, Mu Ku Chen, Tsung Lin Chung, Yu Han Chen, Hsin Yu Kuo, Jia-Wern Chen, Ji Chen, Yi-Teng Huang, Jung-Hsi Wang, Cheng Hung Chu, Pin Chieh Wu, Tao Li, Zhenlin Wang, Shining Zhu, and Din Ping Tsai. Achromatic metalens array for full-colour light-field imaging. *Nat. Nanotechnol.*, 14(3):227–231, March 2019.
- [12] Haoran Ren, Jaehyuck Jang, Chenhao Li, Andreas Aigner, Malte Plidschun, Jisoo Kim, Junsuk Rho, Markus A Schmidt, and Stefan A Maier. An achromatic metafiber for focusing and imaging across the entire telecommunication range. *Nat. Commun.*, 13(1):4183, July 2022.

- [13] J. E. Fröch, L. C. Huang, Q. A. A. Tanguy, S. Colburn, A. L. Zhan, A. Ravagli, E. J. Seibel, K. F. Böhringer, and A. Majumdar. Real time full-color imaging in a meta-optical fiber endoscope. *Elight*, 3(1), 2023. ISSN 2097-1710. doi: ARTN1310.1186/s43593-023-00044-4. URL <GotoISI>://WOS:001094539100001.
- [14] Y. Liu, Q. Y. Yu, Z. M. Chen, H. Y. Qiu, R. Chen, S. J. Jiang, X. T. He, F. L. Zhao, and J. W. Dong. Meta-objective with sub-micrometer resolution for microendoscopes. *Photonics Research*, 9(2):106–115, 2021. ISSN 2327-9125. doi: 10.1364/Prj.406197. URL HYPERLINK"http://gateway.isiknowledge.com/gateway/Gateway.cgi?GWVersion=2&SrcAuth=ResearchSoft&SrcApp=EndNote&DestLinkType=FullRecord&DestApp=WOS&KeyUT=000614927100007"<GotoISI>://WOS:000614927100007.
- [15] Y. Luo, C. H. Chu, S. Vyas, H. Y. Kuo, Y. H. Chia, M. K. Chen, X. Shi, T. Tanaka, H. Misawa, Y. Y. Huang, and D. P. Tsai. Varifocal metalens for optical sectioning fluorescence microscopy. *Nano Letters*, 21(12):5133–5142, 2021. ISSN 1530-6984. doi: 10.1021/acs.nanolett.1c01114. URL <GotoISI>://WOS:000668003400031.
- [16] D. Q. Sun, Y. J. Yang, S. J. Liu, Y. Li, M. Y. Luo, X. L. Qi, and Z. G. Ma. Excitation and emission dual-wavelength confocal metalens designed directly in the biological tissue environment for two-photon micro-endoscopy. *Biomedical Optics Express*, 11(8):4408–4418, 2020. ISSN 2156-7085. doi: 10.1364/Boe.395539. URL <GotoISI>://WOS:000577451600025.
- [17] Yuan Luo, Ming Lun Tseng, Sunil Vyas, Ting-Yu Hsieh, Jui-Ching Wu, Shang-Yang Chen, Hsiao-Fang Peng, Vin-Cent Su, Tzu-Ting Huang, Hsin Yu Kuo, Cheng Hung Chu, Mu Ku Chen, Jia-Wern Chen, Yu-Chun Chen, Kuang-Yuh Huang, Chieh-Hsiung Kuan, Xu Shi, Hiroaki Misawa, and Din Ping Tsai. Meta-lens light-sheet fluorescence microscopy for *in vivo* imaging. *Nanophotonics*, 11(9):1949–1959, May 2022.
- [18] Yu-Hsin Chia, Wei-Hao Liao, Sunil Vyas, Cheng Hung Chu, Takeshi Yamaguchi, Xiaoyuan Liu, Takuo Tanaka, Yi-You Huang, Mu Ku Chen, Wen-Shiang Chen, Din Ping Tsai, and Yuan Luo. In vivo intelligent fluorescence endo-microscopy by varifocal meta-device and deep learning. *Adv. Sci. (Weinh.)*, 11(20):e2307837, May 2024.
- [19] H. Pahlevaninezhad, M. Khorasaninejad, Y. W. Huang, Z. J. Shi, L. P. Hariri, D. C. Adams, V. Ding, A. Zhu, C. W. Qiu, F. Capasso, and M. J. Suter. Nano-optic endoscope for high-resolution optical coherence tomography in vivo. *Nature Photonics*, 12(9):540–+, 2018. ISSN 1749-4885. doi: 10.1038/s41566-018-0224-2. URL <GotoISI>://WOS:000443022200013.
- [20] Ningzhi Xie, Matthew D. Carson, Johannes E. Fröch, Arka Majumdar, Eric J. Seibel, and Karl F. Böhringer. Large field-of-view short-wave infrared metalens for scanning fiber endoscopy. *Journal of Biomedical Optics*, 28(9):094802, 2023. doi: 10.1117/1.JBO.28.9.094802. URL <https://doi.org/10.1117/1.JBO.28.9.094802>.
- [21] Abhijith Rajiv, Yaxuan Zhou, Jeremy Ridge, Per G Reinhall, and Eric J Seibel. Electromechanical model-based design and testing of fiber scanners for endoscopy. *J. Med. Device.*, 12(4):041003, December 2018.

Reply to the Reviewers

Re: Manuscript ID COMMS-24-0224

“Large field-of-view polychromatic metalens for full-color scanning fiber endoscopy: Rebuttal letter for journal reviews”

First, we would like to thank the editor and reviewers for their time and effort to carefully reading our paper and providing us with constructive criticisms. Their comments have surely helped us improve the manuscript. Below you can find the changes made to the manuscript and our replies to all of the reviewers' remarks. We also have changed the manuscript accordingly. The changes are marked with strikethrough line and blue text color.

Reviewer #1, comment #1

Regarding the efficiency result of the metalens, the collimation efficiency falls below 5% at larger deflection angles. Under such conditions, can the imaging performance still be maintained at a satisfactory level?

Our response #1.1

While we admit that the low efficiency at large deflection angles cause the significant reduction of the image contrast, as we stated in section 2.4 in the manuscript, we find that the pattern is still clearly recognizable, as can be seen in Fig.4c. This is because the noise of the image system still remains well below the contrast between the bright and dark blocks of the patterns. Therefore, the suppression of the noise ensures the imaging performance still be maintained at a satisfactory level even with the low efficiency of the lens.

Reviewer #1, comment #2

In the final validation experiment about imaging checkboard patterns by metalens. The author mentions that deconvolution can be applied to remove blurry shadow effects caused by sidelobes. If the collimation efficiency at larger angles is significantly lower than at smaller angles, indicating larger sidelobes, is it still feasible to use deconvolution to effectively correct these artifacts in grayscale images under such conditions?

Our response #1.2

We believe it is still feasible to use deconvolution to correct the artifacts even if the point spread function has position-dependent large sidelobes. Specifically, we find the Richardson–Lucy deconvolution algorithm suitable for this case where the point spread function is position-dependent. We revised the last paragraph of section 2.4 in our manuscript accordingly: "...These artifacts could be mitigated by image deconvolution, which takes into account the angular intensity distribution of the steered beam. Specifically, the Richardson–Lucy deconvolution algorithm is a promising solution as it can accommodate for the position-dependent point spread function of our metalens. "

Reviewer #1, comment #3

The meta-atoms are designed with a cross-shaped geometry, but in the SEM image shown in Fig. 3(b), this shape does not appear to match the intended design. The fabrication process is based on electron beam lithography (EBL). Could the disparity between the designed model and the fabricated structure of the meta-atoms be attributed to the aspect ratio between the structural parameters a and b , resulting in the four corner regions of the cross being too small to fabricate? The paper mentioned that the cross-shaped meta-atoms were intended to help address chromatic aberrations, but how might the incomplete realization of this structure impact the accuracy of the experimental results?

Our response #1.3

We thank the reviewer for pointing out the problem in the SEM image that the shape of the meta-atom does not appear to be cross-shaped. This is because the SEM shows a region in the metalens where the meta-atoms happen to have a very similar structural parameters a and b , which cause them not to look like a cross. We have re-selected an SEM image where the cross-shape of the meta-atoms is much more apparent, and replaced the previous SEM image with it.

We also admit that the corners of the meta-atom are still rounded and do not completely match the intended design. This discrepancy is unavoidable, and could be one of the main reasons for the lower efficiency in experiment compared to simulation. We are working on improving the inverse design framework so that the design could be less sensitive to these fabrication imperfections.

Reviewer #1, comment #4

For the proposed the inverse design of the metalens, could the calculation of the loss function be explained in more detail? The paper notes that the contrast in the image is not discernible and even has the worst efficiency at the blue wavelength. Could this be due to insufficient phase compensation by the lens at this wavelength? Additionally, when using the deep neural network (DNN) to search for the optimal solution, how are the parameters that define the relationship between the RGB phase responses determined? Please clarify

the distribution of the 5673 training samples, particularly the proportion of simulation results at different wavelengths, as the supplementary material suggests that the data distribution may not be uniform.

Our response #1.4

We agree that the definition of the loss function should be explained in more detail, and we revised the section 4.2 accordingly: "...This optimization is performed via using the gradient decent method to minimize a loss function, which is defined as:

$$L = - \sum_{\lambda, \theta} \log(\text{SR}_{\lambda, \theta}) \quad (\text{S1})$$

where the summation is performed over seven beam steering angles θ and three wavelengths λ . Here $\text{SR}_{\lambda, \theta}$ is the Strehl ratio of the incident beams at θ and λ . ~~calculated from the intensity distribution $I_{\lambda, \theta}(x, y)$ on the focal plane.~~ The Strehl ratio that defines the focusing efficiency of the lens is calculated as the maximal value of the simulated light intensity distribution $I_{\lambda, \theta}(x, y)$ on the focal plane divided by the peak intensity on the focal plane of an ideal lens.

The contrast in the image is enough to clearly identify the pattern, although not as high as the refractive lens. This is due to the insufficient coverage of the phase space by the meta-atom library, such that the ideal phase profile of the lens cannot be realized.

We want to clarify that the DNN is not used to directly search for the optimal solution, but to create a meta-model to map the meta-atom to phase and amplitude response, as we stated in the section 4.2 of the manuscript: "... a differentiable function of the meta-atom's phase retardations and amplitude modulations on the incident light with regard to its geometric parameters is essential. This function is approximated by a deep neural network (DNN) meta-model". The parameters in the meta-model that define the phase responses of the meta-atoms at RGB wavelengths are determined through a training process, where the 5673 training samples are all FDTD simulation results of the meta-atom phase and amplitude response at the 3 wavelengths, as we stated also in the section 4.2: "To train this DNN meta-model, 5673 training samples of the meta-atoms' phase and amplitude responses are calculated via Finite-Difference Time-Domain (FDTD) simulations with monochromatic plane wave incident and periodic boundary conditions." The distribution of these phase responses is plotted in the Fig.S4 in the supplementary materials.

Reviewer #1, comment #5

The paper mentions that the camera can only capture a $5^\circ \times 10^\circ$ angular range and multiple scenes must be stitched together to cover a 30° angular range. Could this stitching method introduce data inaccuracies or discontinuities that might affect the precision of the angular intensity distribution measurements?

Our response #1.5

We captured a slightly larger area than $5^\circ \times 10^\circ$ such that the multiple scenes have some overlap region. When stitching the figure together, we use these overlap regions to account for the inaccuracy of the position of the sensor, and ensure no significant discontinuities.

Reviewer #1, comment #6

In the experiment measuring the angular intensity distribution, the light sensor is rotated along a specified trajectory. Please explain how the sensor's rotation trajectory is calibrated to ensure that the light intensity signal is accurately captured at every angle.

Our response #1.6

Our sensor was mounted on a rotational stage, which has angle marks that indicates the rotation angle of the stage. We manually rotated the stage to $0^\circ, 5^\circ \dots 30^\circ$ angles and then took images to capture the light intensity signal around these angles. We admitted that there will be inaccuracies of rotation angles due to the manual operation, however, we captured a slightly larger than $5^\circ \times 10^\circ$ for each image such that the multiple images have overlap region, and used these overlap region to correct for the positional error from manual operations. The scaling of the image (pixel per angle) is essential for converting the image into angular intensity distribution and performing position correction. This scaling was calibrated before the measurement of the angular intensity distribution.

Reviewer #1, comment #7

The proposed polychromatic metalens has issues with efficiency and contrast compared to the monochromatic metalens. While you suggested using post-processing methods to enhance image quality, could you explain if there are possible adjustments in the design methodology that could make the polychromatic metalens more suitable for endoscopic imaging applications?

Our response #1.7

In addition to post-processing methods, we also proposed two ways of improving the image quality of the polychromatic metalens for endoscopic imaging in the manuscript. The first way is using higher index material or taller meta-atoms, which in simulation can improve the lens efficiencies, as we described in the last paragraph of the section 2.2. The second way is implementing a double cladding fiber and confocal configuration setup, as we described in the section 2.4:

"Alternatively, these artifacts could be reduced using a confocal SFE setup (1), in which the scanning fiber not only emits the light for illumination, but also collects the back-scattered light that is re-focused by the metalens into the fiber. The confocal setup can spatially filter out the light not being collimated at the designed position, which greatly reduces the effect of the spurious side beams. Specifically, in conjunction with a double-clad fiber (2) with a single mode core for emission and a larger multimodal clad for collecting return light, the lower collection efficiencies of a confocal setup which would otherwise impede such a device could be mitigated."

Reviewer #1, comment #8

In Supplementary Figure 6, the polychromatic RGB metalens shows beam profiles for five different wavelengths. The beam profiles for 485 nm and 585 nm are larger, and noticeable ghosting occurs at an angle of 15°. Even at the design wavelength of 444 nm, ghosting is present. Could you explain the occurrence of these phenomena in the experimental results?

Our response #1.8

We want to clarify that 485nm and 585nm are not the designed wavelenths of our polychromatic lens. Therefore, the light at these two wavelengths are not supposed to be collimated, which leads to a larger beam size and significant ghosting. Even at the design wavelengths of 643 nm (there is a mistake in the label of the wavelengths in this figure. We have corrected this.), ghosting is present due to the imperfect phase profile of the polychromatic metalens. This ghosting is the main contributor to the lower efficiency of a polychromatic metalens than the monochromatic one.

Reviewer #1, comment #9

When the fiber illuminates along the spiral trajectory (x, y) , according to the description in Section 4.5, if the fiber tip is adjusted by a piezo tube, the incident beam should form an angle and not simply shift parallelly up and down as in the simulation results. Could you provide an explanation for the differences between the light source conditions in the experiment and the simulation?

Our response #1.9

We admit that incident beams emitted from a scanning fiber tip actuated by a piezo tube at different positions do form angles relative to the optical axis instead of simply shifting parallelly up and down, and thank the reviewer for pointing out this unclarified inconsistency. We here clarify that these beam emitting angles are fairly small. We actually simulated both cases (beam emitted with angles and parallelly) and found that there is no significant difference in the beam intensity distributions of these two cases across the whole field-of-view. We added the beam emitting angles to the Tab.S1 and revised the Fig.S1 in the supplementary materials, and revised the section 2.3 in the manuscript accordingly to elaborate this:

"as the tip of the SMF is **parallelly** translated along the tangential axis, the light coupled from the fiber tip is steered by the meta-optics, essentially emulating the scanning mechanism along the tangential axis. **While for the scanning fiber tip, the tip positions have offset from the $z = 0$ plane, and the emitted beams form angles relative to the optical axis instead of simply shifting parallelly along the tangential axis, the position offsets and the beam emitting angles are relatively small compared to the focal lengths of the metalens and the beam steering angle after transmitting through the metalens, as can be seen in the Tab.S1 in the supplementary materials. We found in the simulation that these small differences between the parallelly**

shifted fiber tip and the scanning fiber tip does not lead to significant difference in the intensity distributions of the steered beams."

Figure S1: (a) Cross-section of the ray tracing simulation of the beam scanning system in SFE. (b) Zoom-in cross-section of the scanning fiber and the metalens in the region enclosed by the dashline in (a). (c) The spot diagram of the images of 7 point sources on the image plane. The dashed lines indicate the beam spots with a radius of the diffraction limited FWHMs.

beam	1	2	3	4	5	6	7
fiber tip position z along optical axis (μ m)	0	-0.16	-0.64	-1.44	-2.56	-4.00	-5.76
fiber tip position y_1 on object plane (μ m)	0	40	80	120	160	200	240
beam position y_2 on metalens (μ m)	0	44	89	133	177	222	266
beam emitting angle α ($^\circ$)	0	0.29	0.64	0.93	1.22	1.58	1.86
beam steering angle θ ($^\circ$)	0	5.7	11.5	17.5	23.6	30	36.9

Table S1: The position of the point source, the beam position, emitting angles, and the beam steering angles in the ray tracing simulation in Zemax.

Reviewer #1, comment #10

The paper mentions that the efficiency of the blue light is relatively lower, and the experimental results confirm this. However, in the simulation results (Fig. 2(f)), it is the green light that shows the lowest collimation efficiency. Moreover, in the experimental results, there is a dip at 5 degrees. Could you explain why this happens?

Our response #1.10

We note that while in the simulation results, the three wavelengths have similar efficiencies, with the green light having a slightly lower efficiency. However, the blue wavelength 444 nm has a significantly lower experimental efficiency compared to red and green wavelengths. This might be due to the fact that at this

wavelength, the ratio of wavelength to the 300 nm pitch of the meta-atom is lower, resulting in a lower sampling rate of the light field by the metalens. We have explained this in the last paragraph of the section 2.3 of the manuscript.

Reviewer #2, comment #1

This revised manuscript can be received. However, the large field-of-view is one of the key innovations of this work, but there are few references related to large FOV metalens design. The authors should cite more works, like reference (1-3) and so on. And there is one detail that the authors should correct, that is, the shape of the complex transmission output should be [2x1] not [3x1] in Fig.S5a.

Our response #2.1

We thank the reviewer for recommending our manuscript to be accepted, and we agree that the design principle of large FOV metalenses reported in various other works have significant similarity compared to our cases, such that we should cite these works. We revised the introduction section of the manuscript accordingly: "However, these polychromatic metalenses only work for near-axis imaging. Large field of view (FOV) metalens can be realized by using an aperture stop to spatially separate the incident light from different angles(3, 4, 5). In this paper, via adopting a similar design principle, we report an inverse designed, large FOV polychromatic metalens as a beam steering lens that can support $\sim 70^\circ$ FOV imaging in a RGB-SFE system."

We also thank the reviewer for pointing out the typo in the Fig.S5a, and we had corrected it accordingly.

References

- [1] E. S. Barhoum, R. S. Johnston, and E. J. Seibel. Optical modeling of an ultrathin scanning fiber endoscope, a preliminary study of confocal versus non-confocal detection. *Optics Express*, 13(19):7548–7562, 2005. ISSN 1094-4087. doi: Doi10.1364/Opex.13.007548. URL <GotoISI>://WOS:000232058200033.
- [2] D. Yelin, B. E. Bouma, S. H. Yun, and G. J. Tearney. Double-clad fiber for endoscopy. *Optics Letters*, 29(20):2408–2410, 2004. ISSN 0146-9592. doi: Doi10.1364/Ol.29.002408. URL <GotoISI>://WOS:000224355700026.
- [3] Mikhail Y. Shalaginov, Sensong An, Fan Yang, Peter Su, Dominika Lyzwa, Anuradha M. Agarwal, Hualiang Zhang, Juejun Hu, and Tian Gu. Single-element diffraction-limited fisheye metalens. *Nano Letters*, 20(10):7429–7437, Oct 2020. ISSN 1530-6984. doi: 10.1021/acs.nanolett.0c02783. URL <https://doi.org/10.1021/acs.nanolett.0c02783>.
- [4] Shilin Luo, Fei Zhang, Xinjian Lu, Ting Xie, Mingbo Pu, Yinghui Guo, Yanqin Wang, and Xiangang Luo. Single-layer metalens for achromatic focusing with wide field of view in the visible range. *Journal of Physics D: Applied Physics*, 55(23):235106, mar 2022. doi: 10.1088/1361-6463/ac58ce. URL <https://dx.doi.org/10.1088/1361-6463/ac58ce>.
- [5] Yan Liu, Wen-Dong Li, Kun-Yuan Xin, Ze-Ming Chen, Zun-Yi Chen, Rui Chen, Xiao-Dong Chen, Fu-Li Zhao, Wei-Shi Zheng, and Jian-Wen Dong. Ultra-wide fov meta-camera with transformer-neural-network color imaging methodology. *Advanced Photonics*, 6(5):056001–056001, 2024.

In this work, the authors proposed an inversed-designed polychromatic metalens with a large field-of-view of 70° , suitable for an RGB scanning fiber endoscope. It is claimed that they obtained a close-to-diffraction-limited 0.5° angular resolution. Compared to traditional metalens, their metalens can eliminate the chromatic aberration by meta-atoms library, DNN meta-model, and gradient decent method among three design wavelengths, shown by simulation and experiment. This research has practical engineering applications. The manuscript has high readability which can be acceptable after minor revision.

I have the following comments/suggestions:

1. According to my understanding, the design approach of the RGB polychromatic metalens for the SFE is as follows: firstly, the initial phase profile at each wavelength of RGB is obtained by using Zemax commercial software with even polynomial, then the initial structure arrangement aiming at minimizing the phase error is screened out based on the SiN cross-shape meta-atom library, and finally the geometric parameters of all the meta-atoms across the entire metalens is optimized by using the DNN meta-model and a gradient decent method to maximize the collimated output for the desired angular range.

However, think more carefully, the whole design scheme can be achieved by just using the simplest traversal method without a neural network to model and proxy such a single-structure two-parameter meta-atom library. So, what are the advantages of the DNN meta-model here? Is it possible to speed up the optimization of structures, especially when the metalens size is further increased to mm or even cm level?

If I understand correctly, this fully-differentiable architecture does not seem to make the most of the DNN's inverse design capabilities, such as directly predicting the one-to-many generation of structures according to the target complex amplitude. The term "inverse design" is not prominent enough in this paper, maybe it is easy to misunderstand that the design process is more inclined to a forward optimization loop.

Also, in the manuscript, there is no comparison between this work and a traditional scanning fiber endoscope in terms of size or imaging performance.

About the above, it is recommended that the authors add the relevant discussion to enhance innovation and persuasiveness.

2. In Section 2.2, Fig. 2a-c showed simulated beam intensity distributions in the y-z plane for beam steer angles only from 0° to 17.5° , but the maximum designed beam steer angle is 35° . Similar problems exist in the experimental results of Fig. 3d in Section 2.3. There are some inconsistencies and the results in the y-z plane for maximum beam steer angles are recommended to add.
3. What is the accuracy of the DNN meta-model used in this work? Can it be demonstrated in the missing Fig. S4 (c) and (d) in Supplementary Information Section III?
4. If possible, in Supplementary Information Section V, the authors can consider adding

corresponding phase response distributions like Fig. S4 (a), which will help the readers to understand the impact of the meta-atoms height and material index on the phase diversity more intuitively.